# X-Hacking: The Threat of Misguided AutoML

**Rahul Sharma** [1] **Sumantrak Mukherjee** [1] **Andrea Šipka** [1] **Eyke Hüllermeier** [2] **Sebastian Vollmer** [1]
**Sergey Redyuk** [1] **David Antony Selby** [1]

## Abstract

Explainable AI (XAI) and interpretable machine learning methods help to build trust in model predictions and derived insights, yet also present a perverse incentive for analysts to manipulate XAI metrics to support pre-specified conclusions. This paper introduces the concept of X-hacking, a form of $p$-hacking applied to XAI metrics such as SHAP values. We show how easily an automated machine learning pipeline can be adapted to exploit *model multiplicity* at scale: searching a Rashomon set of 'defensible' models with similar predictive performance to find a desired explanation. We formulate the trade-off between explanation and accuracy as a multi-objective optimisation problem, and illustrate empirically on familiar real-world datasets that, on average, Bayesian optimisation accelerates X-hacking 3-fold for features susceptible to it, versus random sampling. We show the vulnerability of a dataset to X-hacking can be determined by information redundancy among features. Finally, we suggest possible methods for detection and prevention, and discuss ethical implications for the credibility and reproducibility of XAI.

## 1. Introduction

Machine learning (ML) models are increasingly integral to decision-making in critical sectors such as healthcare, criminal justice and public policy. As these models grow more complex, so does the challenge of interpreting the rationale behind their predictions. This has given rise to explainable AI (XAI) methods, which aim to make model reasoning more transparent and globally maintain trust in ML systems.

However, the growing demand for interpretable models and 'data-driven' decisions creates an incentive for actors, unscrupulously—or through lack of time or experience—to seek out model explanations that support pre-specified conclusions, conceal hidden agendas or evade ethical scrutiny.

In this paper, we introduce the concept of explanation hacking, or X-hacking—a form of $p$-hacking (Head et al., 2015) applied to XAI metrics. X-hacking refers to the practice of deliberately searching for and selecting models that produce a desired explanation while maintaining 'acceptable' accuracy. Unlike other adversarial XAI attacks, X-hacking explores plausible combinations of analysis decisions to build a pipeline that might otherwise have been found innocuously. It is a strategy of lying through omission, exploiting a phenomenon known as model multiplicity (Black et al., 2022), where different models offer equivalent predictive performance (Brunet et al., 2022).

We extend existing work on model multiplicity to incorporate the full analysis pipeline, including data preprocessing, feature extraction, choice of model class and hyperparameter tuning, steps which are often poorly reported in scientific publications.

Though the search for such decision sets by hand is long and laborious, it can be accelerated through the exploitation of automated machine learning (AutoML) solutions, or any automated model selection tools that facilitate discovery of a model that may also be found—or presented as found—via human decision-making. Recently, a number of AutoML solutions have been introduced that integrate data preprocessing into the decision-making pipeline, opening up additional avenues for explanation attacks (Salehin et al., 2024).

We demonstrate empirically in a *post-hoc* manner how off-the-shelf AutoML pipelines can be used, even with a limited computational budget, to perform X-hacking on SHAP values for familiar real-world datasets, by 'cherry picking' those models that support a desired narrative. Secondly, we develop a custom AutoML solution to facilitate *ad-hoc* X-hacking at scale, formulating the task as a multi-objective optimisation (MOO) problem, selecting from the Pareto frontier of predictive performance and model explanations to find a 'defensible' model in the *Rashomon set*. We com-

*Equal contribution [1] Deutsches Forschungszentrum für Künstliche Intelligenz GmbH [2] Institute of Informatics, Ludwig-Maximilians-Universität München. Correspondence to: David Antony Selby <david.antony_selby@dfki.de>.

*Proceedings of the 42nd International Conference on Machine Learning*, Vancouver, Canada. PMLR 267, 2025. Copyright 2025 by the author(s).

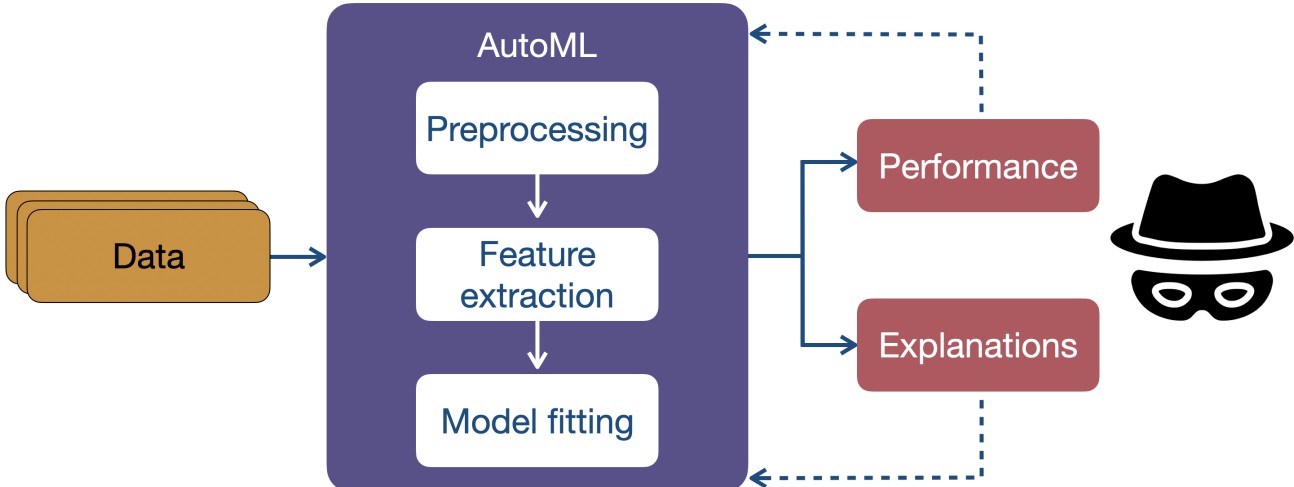

Figure 1: Framework for 'X-hacking' using AutoML. For a given dataset, an automated pipeline evaluates different combinations of modelling decisions, simultaneously optimising performance and XAI metrics

pare random and Bayesian hyperparameter optimisation to maximise accuracy and minimise SHAP explanations of top features from a common baseline and show that Bayesian hyperparameter optimisation is, on average, three times faster in finding defensible models. We compare the time taken in *post-hoc* and *ad-hoc* settings to obtain a defensible model for susceptible datasets: the *ad-hoc* framework took an average of 6 minutes—10 times faster than *post-hoc*.

Our contributions are as follows: (1) we show the ease with which X-hacking can be performed using off-the-shelf software with attacks on different types of SHAP summaries in a *post-hoc* manner; (2) we perform directed (*ad-hoc*) X-hacking at scale, using Bayesian multi-objective optimisation, comparing it with random sampling; (3) we demonstrate how some datasets are more vulnerable than others to confirmatory explanations unrepresentative of the ground truth; (4) we compare *post-hoc* and *ad-hoc* X-hacking by calculating the time required to find the first defensible model; (5) we discuss methods for prevention and detection of X-hacking. Code to replicate our experiments is available on GitHub.

## 2. Background

In this section, we introduce the key concepts of XAI, multiplicity and questionable research practices, whose intersection gives rise to the threat of X-hacking.

**Explainable AI**  Traditional statistical models, such as generalized linear regression models or decision trees, are inherently or *ante-hoc* explainable and easily interpreted by humans. Algorithmic ML models (see Breiman, 2001) tend to be more opaque, requiring *post-hoc* explanation methods

to examine the effect of different inputs and investigate potential biases (Gunning et al., 2019). These methods can be model-specific, such as feature importance for random forests and DeepLIFT for deep learning (Shrikumar et al., 2019), or model-agnostic, such as LIME (Ribeiro et al., 2016) and SHAP values (Lundberg & Lee, 2017). While not without criticisms (Gosiewska & Biecek, 2020), Shapley value-derived explanations enjoy widespread adoption (Hickling et al., 2023). Increasing interest in XAI and algorithmic fairness (Garg et al., 2020), and their inclusion in future legislative frameworks regulating the use of AI (Bordt et al., 2022), brings incentives for manipulation. Adversarial attacks against XAI or fairness metrics are an emerging area of research. Attacks can involve manipulating a model's architecture or its explanations or 'poisoning' training data (Baniecki & Biecek, 2024). In particular, *fairwashing* describes the practice of attempting to deceive measures of algorithmic fairness. Approaches include approximating a complex, biased model with an ostensibly fair, interpretable one (Aivodji et al., 2019), designing an unfair model that switches to generating 'fair' predictions when being audited, akin to automotive manufacturers cheating emissions tests (Slack et al., 2020), or performing biased sampling of the data points used to compute the SHAP values (Laberge et al., 2023a). Recently, authors have proposed methods to detect and thwart such attacks (Shamsabadi et al., 2022; Carmichael & Scheirer, 2023).

**Multiplicity**  Models and their explanations are not unique; multiple alternative models can deliver roughly equivalent predictive performance (Pawelczyk et al., 2020; Brunet et al., 2022; Rudin et al., 2024). This phenomenon is known variously as model multiplicity (Marx et al., 2020), underspecification (D'Amour et al., 2022) or the Rashomon

effect (Breiman, 2001; D'Amour, 2021). Procedural multiplicity describes models that have identical predictive accuracy but differ in their internal structures (Mehrer et al., 2020; Black E., 2021), a concept also known as a *Rashomon set* (Fisher et al., 2019) or the 'set of good models' (Ganesh et al., 2025). A special case of procedural multiplicity is predictive multiplicity: where models achieve the same accuracy but produce different predictions (Black et al., 2022). The effects of multiplicity can be felt at the level of individual predictions (Marx et al., 2020), and on global properties such as model fairness and robustness (Rodolfa et al., 2021; D'Amour et al., 2022). Recent work has studied multiplicity in specific classes, including linear and generalized additive models (Dong & Rudin, 2020; Zhong et al.), neural networks (Mehrer et al., 2020; Black E., 2021; Laberge et al., 2023b), sparse decision trees (Wang et al., 2022) and random forests (Smith et al., 2020).

*p*-**hacking** In null hypothesis significance testing, *p*-hacking is a questionable research practice whereby researchers, equipped with many possible data analysis choices, only report those yielding a 'significant' result (Wasserstein & Lazar, 2016) while others languish unpublished (Scargle, 1999). Strategies for *p*-hacking include multiple testing, selective reporting and favourable imputation (Stefan & Schönbrodt, 2023). Any choices that are contingent on data, rather than a pre-specified study protocol, are vulnerable to these 'researcher degrees of freedom', whether or not there is a conscious desire to mislead (Gelman & Loken, 2013). This phenomenon is considered to have contributed to the reproducibility crisis in scientific research (Wicherts et al., 2016). Even in systematic reviews of literature, *p*-hacking may not be easy to detect (Rooprai et al., 2022), though pre-registrations and replication studies aim to discourage the practice (Hussey, 2021). In quantitative sciences, details of data preparation are often poorly reported, yet can have a significant impact on results (Jani et al., 2023). Some authors (Heyard & Held, 2022) suggest data and code should be made available at the preprint stage rather than at final publication time, however the willingness of researchers to do so varies by discipline (Hussey, 2023; Goldacre et al., 2019).

## 3. Problem statement

The goal of explanation hacking is to find a model that corroborates a predetermined view of the world. For example, a pharmaceutical company may wish to show that a drug is not associated with adverse outcomes, an organisation may wish to claim decisions were not biased against a protected group, or a lobbyist may wish to show that smoking protects against cancer.

Thanks to predictive multiplicity, automated model selection

tools (including AutoML) make it easy for people to obtain models with desirable explanations—deliberately or not. Existing studies of multiplicity, however, are restricted to specific model families and do not cover modern end-to-end data science pipelines.

Consider a set of models $\mathcal{M}$ designed to perform a task (e.g. classification) on a dataset $D$. We can quantitatively evaluate a model $m \in \mathcal{M}$ via a procedure $\text{eval}(m, D, Q)$, where $Q$ is a quality measure, either a predictive performance metric (e.g. accuracy), denoted $Q_D(m) = \text{perf}(m)$; or an inferential summary for a feature of interest $Q_D(m) = \mathcal{I}(m, \mathbf{x})$ for $\mathbf{x} \in D$. Classical inference is typically tied to measures such as model coefficients, *p*-values or effect sizes (e.g. Cohen's $d$), which have capacity to mislead (Pogrow, 2019), but also limit the family of models and data science pipelines at the (unscrupulous) analyst's disposal. Model-agnostic XAI metrics, such as SHAP values, open up a wider range of algorithms and possible combinations of hyperparameters that may yield varied interpretations.

Ordinarily, model search is an optimisation problem,

$$\underset{m \in M}{\arg\max}\, Q_D(m), \tag{1}$$

where $Q = \text{perf}(m)$. X-hacking extends this with a competing objective, $\mathcal{I}(m, \mathbf{x})$.

A *post-hoc* strategy retrieves a (Rashomon) set of 'good' models exceeding a pre-specified baseline, e.g. $M_s = \{m : \text{perf}(m) \geq b\}$, then optimises the other metric, e.g. $\arg\max_{m \in M_s} \mathcal{I}_D(m, \mathbf{x})$. Alternatively, an *ad-hoc* strategy combines these metrics, *inter alia* (e.g. risk of getting caught, $\mathcal{Z}$) into a multi-objective optimisation, where $\mathbf{Q}_D = (q_1, \ldots, q_n)$ is a vector.

Here, we intentionally do not specify the expressiveness of conditions that encode model interpretations or the process of generating the data science pipeline.

## 4. Explanation Hacking

In this section, we outline a framework, illustrated in Figure 1, that unscrupulous or inexperienced actors might employ to find publishable ML models with predetermined, possibly contrarian explanations. We consider three quantifiable metrics that an actor might consider when trying to hack XAI metrics.

**Explanations** SHAP values are typically presented visually in the form of force plots, SHAP summary plots, partial dependence graphs or feature importance (see, e.g. Lundberg et al., 2018; Stenwig et al., 2022). If a specific data point, $x_i$, such as an individual patient or credit card applicant (the 'suing set'; Shamsabadi et al., 2022), is of interest, then the adversary can 'fish' for a trained model that gives

a SHAP value in the desired direction, e.g. $\mathcal{I}(m, x_i) \to 0$, allowing them to claim that an AI-assisted decision was *not* based on a protected characteristic. Model- or large group-level explanations require aggregated SHAP statistics. If interpreting a model based on ranked feature importance, then one could set $\mathcal{I}(m, \mathbf{x}) = \text{rank}_D(\text{mean}(|\text{SHAP}(\mathbf{x})|)) > k$, to seek models where the feature of interest is not in the 'top $k$' most important features. Partial dependence plots, despite being a visual summary, may also be 'hacked': an adversary can examine the relationship between SHAP and a specific feature approximately with the slope of a simple linear model (Figure 4), or even $\mathcal{I} = \text{Corr}(\mathbf{x}, \text{SHAP}(\mathbf{x}))$ and search for models with positive, negative or no correlation.

**Defensible models** The premise of an X-hacking framework is for an agent to select combinations of analysis decisions—such as the model family, architecture, data transformations or imputation methods—each of which is individually *defensible* (i.e. could be justified in peer review), but whose combination leads to a confirmatory result. A decision is defensible if a plausible justification can be proffered in the report accompanying the analysis. For this we consider two alternative appeals to authority: the model has superior predictive performance to a baseline (i.e. the Rashomon set of 'good models' Ganesh et al., 2025); or the choice is considered 'standard practice' by peers in the target domain. As the former is more readily quantified (i.e. $\text{perf}(m)$) in an automated search, for the remainder of this paper we use the term 'defensible' models simply to refer to the Rashomon set of models with good predictive performance.

Hence, we treat X-hacking as balancing competing objectives of predetermined explanations, measured via XAI metrics, and predictive performance.

### 4.1. Cherry Picking

Let $m_\downarrow$ be a suitable baseline model and let $b = \text{perf}_D(m_\downarrow)$ be its performance, and $\mathcal{I}_k(\mathbf{x}) = \text{mean}(|\text{SHAP}_m(\mathbf{x})|)$, the mean absolute SHAP of feature $\mathbf{x}$ for model $m$. Let $\mathbf{x}^*_{m_\downarrow}$ be the top ranking feature by importance ($\text{rank}_{m_\downarrow}(\mathbf{x}^*_{m_\downarrow}) = 1$) for the baseline model, where $\mathbf{x}^*_{m_\downarrow} := \arg\max_{\mathbf{x} \in D} \mathcal{I}_{m_\downarrow}(\mathbf{x})$. Let $M$ be the search space of models in the AutoML solution and $S_{t,D} \subset M$ the set of models returned by AutoML after running for duration $t$. The Rashomon set of models having comparative or superior performance to the baseline is given by $\mathcal{R}_{t,D} := \{m \mid m \in S_{t,D} \wedge \text{perf}_D(m) \geq b\}$. Let $\mathcal{C}_{t,D}$ be the set of models that confirm the desired hypothesis, e.g. $\mathcal{C}_{t,D} = \{m \mid \text{rank}_m(\mathbf{x}^*_{m_\downarrow}) > k\}$, the set of models where $\mathbf{x}^*_{m_\downarrow}$ is not in the top $k > 1$ most important features.

A human or machine can then simply 'cherry-pick' any model from $\mathcal{C}_{t,D} \cap \mathcal{R}_{t,D}$, the intersection of the set of models

that confirm the desired hypothesis and the Rashomon set: the *confirmatory Rashomon set*.

### 4.2. Directed Search

A more efficient, but higher-effort, *ad-hoc* approach involves custom multi-objective optimisation.

Let $M$ be the population of models in the AutoML search space and $\mathbf{Q}_D(m, \mathbf{x}) = \{q_1, q_2\}$ be a vector of objectives, with $q_1 = -\text{perf}(m, D)$ and $q_2 = \mathcal{I}_k(m, \mathbf{x})$ for $m \in M$. Then, for each feature of interest $\mathbf{x}$, we select the optimal model using (1), either from a random subset of models $\{m_j\}$ or from the suggestions of a Bayesian optimiser.

We say model $m$ (Pareto)-*dominates* another model $m' \in \Lambda$ for a feature $\mathbf{x}$ if and only if there is no criterion $q_i$ in which $m' \succ m$ ($m'$ is superior to $m$), and at least one criterion $q_j$ in which $m \succ m'$. An analyst may select a trained model from the *Pareto front* of feasible solutions.

**Remark** A weighted-sum scalarisation approach to multi-objective optimisation can also be used. Details are in the Appendix C.

## 5. Experiments

In this section, we explain our experimental setup and present empirical results of X-hacking as described in § 4.1 and § 4.2. Next, we demonstrate how certain features can be robust to X-hacking. Finally, we compare X-hacking methods to see how quickly they are each able to find a model that is in the confirmatory Rashomon set.

### 5.1. Cherry Picking

To demonstrate the feasibility of 'easy' X-hacking approaches, we present an empirical evaluation using off-the-shelf XAI and AutoML software packages and several familiar real-world datasets.

**Data** We selected 23 datasets from the OpenML-CC18 classification benchmark (Bischl et al., 2021), where the task was binary classification from tabular features. For all datasets, categorical features were one-hot encoded and samples with missing values omitted.

**Tools** Python packages `scikit-learn` (Pedregosa et al., 2011) and `auto-sklearn` (Feurer et al., 2019) built and trained the ML models, and package `shap` (Lundberg et al., 2018) estimated SHAP values from the successfully evaluated models. Models were evaluated on an internal cluster using 192 CPUs with 300GB RAM.

**X-hacking** The baseline model was a random forest classifier trained with `scikit-learn` with all parameters set to

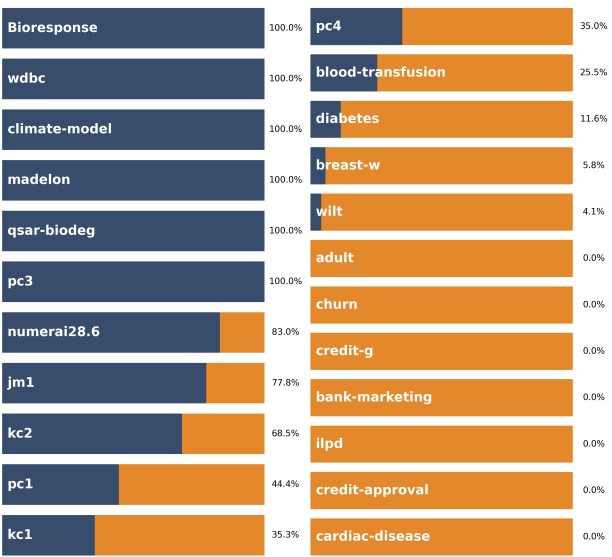

Figure 2: Proportion of models with superior predictive performance where 'top' feature from baseline is no longer ranked in the 'top 3'. Each bar represents one dataset

default. To get different candidate models against the baseline, we ran `auto-sklearn` for one hour on each dataset, with a runtime limit of 100 seconds for each model configuration. For the baseline and all models evaluated in AutoML, the `shap`'s model-agnostic `KernelExplainer` routine was employed to compute SHAP values, using a background sample size 50 and test sample size 100.

Manipulation of two different SHAP-based explanations was attempted: feature importance ranking and partial dependence plots. Feature importance was determined as either the ranking or magnitude of the mean absolute SHAP values of the features. The visual appearance of the feature dependence plot was quantified by a simple linear regression model of SHAP against respective feature values, the

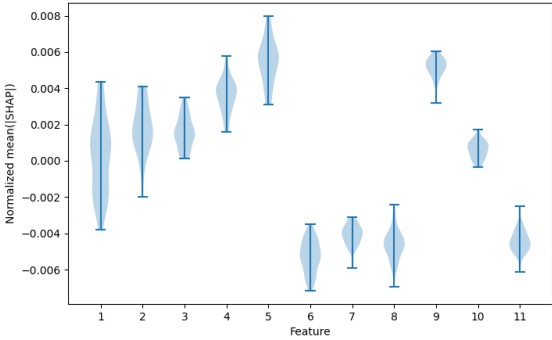

Figure 3: Changes in feature importance relative to baseline for `cardiac-disease` data. Features are ordered by baseline importance.

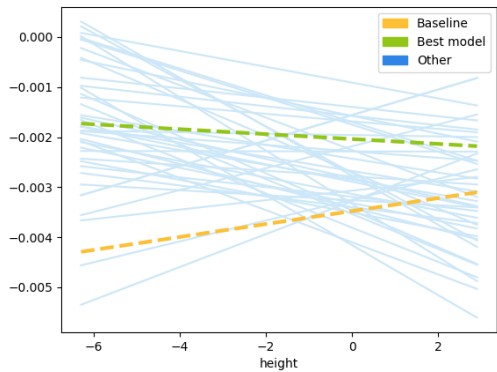

Figure 4: Lines of best fit of SHAP against feature `height`, for different models evaluated by AutoML indicating availability of several choices for models (Other) that 'flip' the SHAP value

slope indicating the sign of any (approximate) linear relationship between the feature and the outcome. Predictive performance was measured by accuracy score; the AutoML pipeline was optimized for this metric as a single objective (the default) with no consideration given to SHAP during the initial model search; model explanations were evaluated *post-hoc*.

**Results: feature importance** With no agenda of our own, we arbitrarily chose the most important feature at baseline as the target for explanation manipulation. Figure 2 shows, for each dataset, the proportion of models returned by the AutoML pipeline where this feature is no longer ranked in the 'top 3' for feature importance, conditional on the model's predictive accuracy being higher than at baseline.

For about a third of the datasets, it was not possible in the time available to find a more accurate ML model that 'toppled' the baseline most important feature. For a similar proportion of datasets, almost every more performant model had different top features. Figure 3 shows the distribution of change in mean absolute SHAP values, compared to the baseline random forest classifier, for the `cardiac-disease` dataset. The distribution of differences is close to zero for the top three features, suggesting that for this dataset, it may be difficult to find a modelling pipeline that renders the 'top' feature at baseline as unimportant in a reported model.

**Results: feature dependence** As well as the importance of a feature—an unsigned quantity—a misguided analyst may wish to manipulate the shape of the effect. The lines of best fit between SHAP and feature `height` (which has high importance at baseline; feature 2 in Figure 3) for the `cardiac-disease` dataset are plotted in Figure 4. Several modelling choices are available that 'flip' the SHAP

value for this feature while being more accurate than the baseline. Hence, an analyst might present the underlying partial dependence plot as evidence of the negative effect of height on cardiac disease—omitting the information that alternative models, including the baseline, show either positive or no linear effect. Further results are given in Appendix D.

Whilst 'cherry picking' a desired model from an undirected set of candidates may seem easier approach to implement, such a 'cherry' may not be found in a reasonable time using standard AutoML tools, or it may be difficult to define a space of uniformly defensible models.

## 5.2. Directed Search

To deliberately hack explanations, we created a custom AutoML solution that allows us to perform multi-objective optimisation of performance and explanations in a distributed manner on familiar real-world datasets.

**Data**   as given in § 5.1

**Tools**   For this experiment, the AutoML solution should support multi-objective optimisation over many model classes and their hyperparameters in a distributed manner. Most AutoML solutions do not support these requirements seamlessly (see Appendix A). Therefore, we create our custom AutoML solution by using the search space from `auto-sklearn`, enabling Bayesian multi-objective optimisation using `optuna` and running of pipelines in a distributed manner using `Ray-tune`. More details are in Appendix B.

**Directed X-hacking**   To demonstrate intentional manipulation of explanations, the top 4 features from each dataset were taken, which correspond to the top 4 most important features in the random forest baseline. In total, we perform this experiment for 92 features. To get different candidate models against the baseline, the custom AutoML setup was run for 12 hours for the top 4 features indicated by the baseline, with a runtime limit of 1 hour for each model configuration suggested by the Bayesian optimiser. For all models evaluated in our custom AutoML setup, `shap`'s model-agnostic explainer, `Kernel Explainer` routine was employed to compute SHAP values, using a background sample size 50 and test sample size 100. Manipulation of SHAP-based feature importance was attempted. MOTPESampler (Ozaki et al., 2022) was used as the Bayesian optimiser. Feature importance was determined as the magnitude of the mean absolute SHAP values of the features. Predictive performance was measured by accuracy score. The AutoML pipeline was optimised for these two objectives simultaneously as a multi-objective optimisation where accuracy score was optimised to be maximum

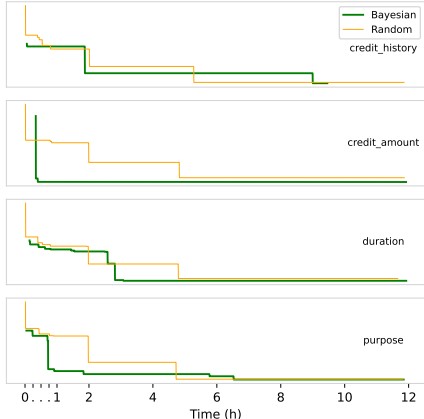

Figure 5: Cumulative minimum of mean absolute SHAP for top 4 features of dataset `credit-g` for Bayesian optimisation and random sampling

and mean absolute SHAP was optimised to be minimum. To demonstrate how fast/slow Bayesian multi-objective optimisation is, we run the setup with same time and resource limits but with a random optimiser.

**Results: directed search**   The cumulative minimum mean absolute SHAP of four 'top' (at baseline) features for defensible models was recorded for random sampling and Bayesian optimisation over 12 hours of runtime and plotted in Figure 5 for the `credit-g` dataset. To summarise this information for all datasets, Figure 6 shows the ratio of duration of time when cum $\min$ mean($|\text{SHAP}(\mathbf{x}_i|)$ for Bayesian optimisation was always lower than random sampling over the top 4 features from baseline across all datasets. For these features, on average, Bayesian optimisation reached the overall minimum of mean absolute SHAP obtained by random sampling 3 hours earlier. For the remaining features where the overall minimum mean absolute SHAP was lower for random sampling, on average, random sampling reached the overall minimum obtained by Bayesian optimisation nearly 55 minutes earlier. This indicates, on average, Bayesian optimisation is 3 times faster than random sampling for a majority of features to defensibly hack explanations. For some features (20/92), no defensible models were found by both random sampling and Bayesian optimisation in the current setting, indicating their robustness to X-hacking.

## 5.3. Vulnerability to X-hacking

Quantifying the vulnerability of a dataset to X-hacking is equivalent to measuring predictive multiplicity: existing metrics include *Rashomon Capacity*, based on the KL-divergence for a set of probabilistic classifiers (Hsu & Calmon, 2022). In practice, predictive multiplicity may only be

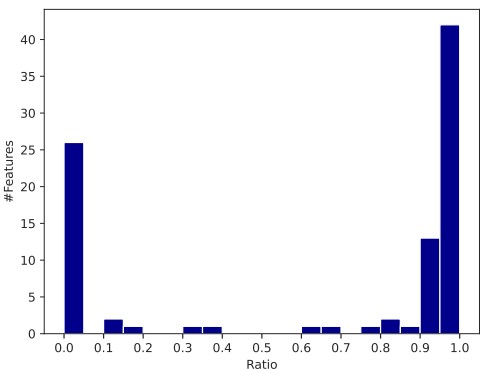

Figure 6: Ratio of time duration (over 12 hours) Bayesian MOO had better optimisation for mean absolute SHAP than random MOO for top 4 features from all datasets

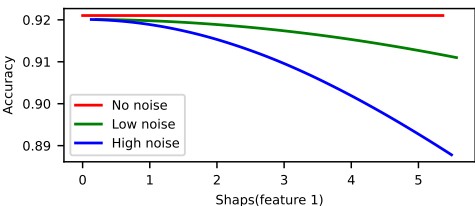

Figure 7: Trade-off between accuracy and (approximate) SHAP dependence slope

approximated, as it is typically computationally infeasible to evaluate every possible analysis pipeline. To determine when a feature is robust to X-hacking, we perform a simulation.

**Simulation** Using the multi-objective optimisation package `Optuna` (Akiba et al., 2019), we evaluate 50 models on a simulated dataset. SHAP values are calculated using the `shap` package's model-agnostic explainer. Sample sizes are the same as in § 4.1.

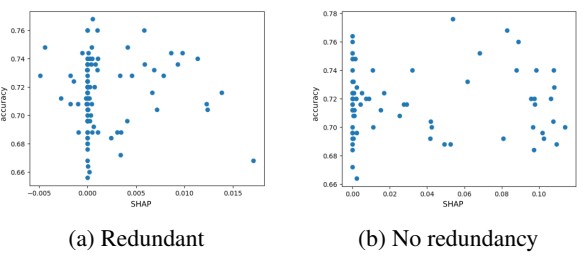

(a) Redundant      (b) No redundancy

Figure 8: Predictive accuracy versus SHAP values for simulated features (each point corresponds to a trained pipeline).

**Simulation results** Figure 7 shows three simulated datasets containing two collinear features measured with varying levels of noise. Additivity of SHAP values allows attribution to be transferred from one feature to the other without loss of accuracy, but this effect decreases with the signal-to-noise ratio. This shows the ability to manipulate XAI metrics for a given feature depends on the level of information shared with others.

We demonstrate in Figure 8 that it is possible to flip SHAP values of features with redundant information present in other variables, without loss of accuracy, but not for variables without such information redundancy. Having access to the data-generating process allows us to be aware of the relations between variables and the redundancy of information present within them. We select two variables (one with redundant information captured by dependent nodes and one without) to demonstrate which variables can and cannot be flipped for a particular dataset. Therefore, how easy it is to manipulate the SHAP values of a particular feature—and hence the need for a directed search or a more tightly curated search space—is determined by redundancy among the features. If all input features are completely independent of one other, manipulation of SHAP values—without change of model predictive accuracy—is not possible.

### 5.4. Cherry-picking vs directed search

As opposed to cherry-picking a model from a Rashomon set obtained after passing a dataset through an off-the-shelf AutoML solution, using directed search, one can continue performing X-hacking until a confirmatory model is found. Using the results obtained in the experiments § 5.1 and § 5.2 we see the time it takes for both X-hacking techniques to find the first confirmatory defensible model in one hour of runtime. In this experiment, we see those defensible models where the rank of the top feature from baseline is no longer in the 'top 3'.

**Data and Tools** as given in § 5.1 and § 5.2

**Results: First confirmatory defensible model** For 15 out of 23 datasets, using directed search, a defensible model could be found without waiting for an hour, as opposed to the case of *post-hoc* setting. For 14 such datasets, a defensible model could be found in less than 6 minutes of directed X-hacking. Table 1 shows the time it took (in seconds) for directed X-hacking to find the first defensible model. This demonstrates on average a 10x faster finding of defensible models for datasets and their features vulnerable to X-hacking.

Table 1: Time taken (in minutes and seconds) by directed search to find the first confirmatory defensible model for the top feature of each dataset according to the baseline random forest in an *ad-hoc* X-hacking setting.

| Dataset Name | Time taken |
| --- | --- |
| breast-w | 3 s |
| ilpd | 5 s |
| diabetes | 7 s |
| blood-transfusion | 10 s |
| cardiac-disease | 24 s |
| pc1 | 37 s |
| kc2 | 42 s |
| wilt | 47 s |
| climate-model | 1 m 4 s |
| wdbc | 1 m 16s |
| numerai28.6 | 1 m 45 s |
| credit-approval | 2 m 17 s |
| pc3 | 4 m 39 s |
| kc1 | 5 m 20 s |
| credit-g | 55 m 18 s |

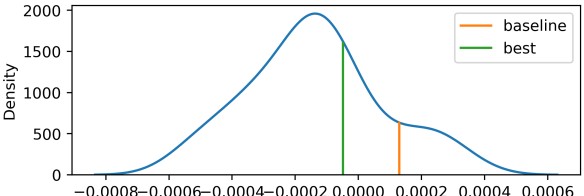

Figure 9: Marginal distribution of SHAP slopes over AutoML-evaluated models for the feature `height`

## 6. Detection and Prevention

Understanding the scale of the problem—as well as detection and prevention—requires systematic ability to identify potentially selectively reported ML pipelines. In this section, we outline several approaches that might be used to highlight inconsistencies indicative of explanation hacking. We consider these approaches to complement existing proposals for detection of fairwashing (e.g. Shamsabadi et al., 2022). Feasibility of these methods depends on the level of access to raw data, code and detailed analysis protocols.

**Explanation histograms**   With access to raw data, a reviewer may set up their own AutoML pipeline to explore the space of possible pipelines, performing the same process as in § 4.1, but with the goal of visualising the marginal distribution of XAI metrics over likely pipelines and locating the reported metric within this distribution. A reported value lying in the extreme tails (or outside the distribution completely) would be consistent with explanation hacking, and may be quantified via hypothesis testing. With infinite time,

computing power and a well-specified search space, the auditor may explore all possible pipelines, but in practice the explored pipelines may not be representative, so an effective search would exploit domain knowledge or additional information. The goal is not necessarily to reverse-engineer the exact pipeline used, rather to investigate the sensitivity of reported explanations. An example is given in Figure 9.

**Pipeline analysis**   How close is a coded pipeline to what is reported in the paper, or to accepted practices in the field? Though a labour-intensive task for a human domain expert, pipeline–pipeline similarity can be converted into a graph isomorphism problem, if data analysis pipelines are represented as directed acyclic graphs and the similarity quantified as graph edit distance (Ono et al., 2021). A challenge of this approach is extracting the pipeline correctly (Redyuk et al., 2022) from code or text. Moreover, parsable, discoverable pipelines in the literature may not be representative of best practice.

## 7. Discussion and limitations

The experiments presented here exploit SHAP values, which have their own limitations (Huang & Marques-Silva, 2023), but the general framework may be adapted to $p$-values or other XAI metric. In principle, any *post-hoc* explanation method susceptible to model multiplicity faces similar risks. A systematic exploration of different XAI metrics is planned as future work.

The unethical use of statistical tools has a long history in science (Huff, 1993), exacerbated by a 'publish or perish' culture. While we do not present any evidence of X-hacking 'in the wild', we argue the latest developments in AutoML and XAI provide means, motive and opportunity. A guided search over an ever-growing field of data science operators, including X-hacking, presents a more dangerous type of scientific 'fake news' as it can be done at scale, with low additional effort by using readily available tools, and it is much more difficult to detect—it does not take much effort to disguise its use even from an educated, expert mind.

With small sample sizes, the effort and statistical expertise needed to 'squeeze' insights out of the data was much higher. AI methods enable such dishonest practices to be (semi-)automated, scalable and accessible to researchers and practitioners at lower effort and cost. The deception need not even involve all authors of a publication, as one author might be tempted to use X-hacking in isolation, without knowledge of their co-authors. Pre-registered analysis plans could prove valuable, if comprehensive enough to cover the range of models and hyperparameters in an AutoML search space.

Aware of the ease of lying through X-hacking, we do not present a comprehensive how-to guide or a tool that

could be used by an adversarial actor, demonstrating only a proof of concept. However, our code and experiments are open source (GitHub: `https://github.com/datasciapps/x-hacking`) and we encourage others to adapt the experimental setting to their own view on what plausible effort might be on the part of an adversary.

In this paper, we have considered the case of inexperienced scientists or adversarial analysts, however the potential for perverse incentives ranges much further, due to additional financial motivations for reporting misleading explanations. Existing and future legislative frameworks regulating the use of AI, particularly in high-stakes scenarios (e.g. EU AI act; GDPR) increasingly mandate or encourage explainability (Panigutti et al., 2023) including in industrial contexts.

We view our demonstration of the plausibility of this new modality of lying as a necessary step for developing effective countermeasures.

## Impact Statement

This paper introduces the concept of 'X-hacking', a misuse of XAI and AutoML tools to produce defensible but misleading model explanations. The study highlights a potential avenue for adversarial misuse that could undermine trust in machine learning systems and exacerbate the reproducibility crisis in scientific research.

While the goal of this work is to inform the community of the risks associated with X-hacking and propose countermeasures, there is a potential for these findings to be misused by actors seeking to exploit these vulnerabilities. Technical details that would lower the barrier to adversarial use have been omitted. Our open source code is intended to facilitate reproducibility and to encourage further research into detecting and preventing X-hacking. We emphasize the importance of responsible dissemination of related tools and frameworks and advocate for the inclusion of detection strategies as part of routine XAI evaluations.

The societal implications of X-hacking are potentially broad, particularly in high-stakes domains such as healthcare, finance and criminal justice, where biased or misleading explanations could have significant ethical and legal ramifications. Our aim is to mitigate these risks through awareness and contribute to the development of more robust and trustworthy AI systems.

## Acknowledgements

The project on which this report is based was funded by the Federal Ministry of Education and Research under the funding code 03ZU1202JA . The responsibility for the content of this publication lies with the author.

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

## A. AutoML and multi-objective optimisation

In order to conduct experiments to answer our research questions, an off-the-shelf AutoML solution should be able to perform multi-objective optimisation with custom objective functions over many model classes. However, to our knowledge, an off-the-shelf solution that incorporates all the required capabilities in a single package does not exist. Table 2 shows different AutoML and hyperparameter optimisation libraries and their capabilities: support for multi-objective optimisation (MOO), support for many model classes (MMC) and support for custom objective functions (CO).

Table 2: Off-the-shelf capabilities of different AutoML and hyperparameter optimisation solutions. Our solution (36) enables multi-objective optimisation in an existing AutoML library

| # | Library | Type | MOO | MMC | CO |
|---|---------|------|-----|-----|-----|
| 1. | Ax (Bakshy et al., 2018) | hyperopt | Yes | No | No |
| 2. | BaYesOpt (Martinez-Cantin, 2014) | hyperopt | No | No | No |
| 3. | BOHB (Falkner et al., 2018) | hyperopt | No | No | No |
| 4. | CFO (Wu et al., 2021) | hyperopt | No | Yes | No |
| 5. | BlendSearch (Wang et al., 2021a) | hyperopt | No | Yes | No |
| 6. | HEBO (Cowen-Rivers et al., 2022) | hyperopt | yes | no | yes |
| 7. | HyperOpt (Bergstra et al., 2013) | hyperopt | No | No | Yes |
| 8. | SkOpt (Head et al., 2021) | hyperopt | No | No | Yes |
| 9. | Nevergrad (Bennet et al., 2021) | hyperopt | No | No | Yes |
| 10. | Optuna (Akiba et al., 2019) | hyperopt | Yes | No | Yes |
| 11. | ZooOpt (Liu et al., 2022) | hyperopt | No | No | Yes |
| 12. | TPOT (Parmentier et al., 2019) | AutoML | limited | Yes | Yes |
| 13. | TPOT2 Alpha (Le et al., 2020) | AutoML | limited | Yes | Yes |
| 14. | Auto-Sklearn (Feurer et al., 2019) | AutoML | No | Yes | Yes |
| 15. | Hyperopt-sklearn (Komer et al., 2019) | AutoML | No | Yes | No |
| 16. | FLAML (Wang et al., 2021b) | AutoML | Yes | Yes | Yes |
| 17. | H2O AutoML (LeDell & Poirier, 2020) | AutoML | No | Yes | No |
| 18. | AutoGluon (Tang et al., 2024) | AutoML | No | Yes | Yes |
| 19. | MLBox (Vasile et al., 2018) | AutoML | No | Yes | No |
| 20. | Auto-Keras (Jin et al., 2019) | AutoML | Yes | No | Yes |
| 21. | AutoGluon-Tabular (Erickson et al., 2020) | AutoML | No | Yes | Yes |
| 22. | AutoWEKA (Thornton et al., 2013) | AutoML | No | Yes | No |
| 23. | AutoML mljar-supervised (Płońska & Płoński, 2021) | AutoML | No | Yes | No |
| 24. | Hyperactive (Simon Blanke, since 2019) | hyperopt | Yes | No | Yes |
| 25. | Optunity (Claesen et al., 2014) | hyperopt | No | No | Yes |
| 26. | HyperparmeterHunter (McGushion, 2018) | hyperopt | No | No | No |
| 27. | KerasTuner (O'Malley et al., 2019) | hyperopt | No | No | Yes |
| 28. | Talos (Autonomio) | hyperopt | No | No | No |
| 29. | ML.Net (Ahmed et al., 2019) | AutoML | No | Yes | No |
| 30. | NNI (Microsoft, 2021) | AutoML | No | No | No |
| 31. | Azure AutoML (Microsoft, 2018) | AutoML | No | Yes | No |
| 32. | Amazon SageMaker (Amazon, 2017) | AutoML | No | Yes | No |
| 33. | Google Vertex AI (Google Cloud) | AutoML | Yes | No | No |
| 34. | Ray Tune (Liaw et al., 2018) | hyperopt | Yes | No | Yes |
| 35. | syne-tune (Salinas et al., 2022) | hyperopt | Yes | no | yes |
| 36. | Ray tune + optuna + autosklearn | Automl | Yes | Yes | Yes |

## B. Custom AutoML solution

To enable distributed multi-objective optimisation (MOO) over the search space of classification models, data pre-processors, and pre-processing steps provided by `autosklearn` we incorporated `optuna` for MOO and `Ray Tune` for distributed computing. We repurposed the search space of `auto-sklearn` and created our own solution to enable multi-objective optimisation over the search space of the models and their hyperparameters provided by `auto-sklearn`. By using `optuna`(Akiba et al., 2019), we added a multi-objective optimiser on top of `auto-sklearn`. To enable distributed computation of models and optimisation of their hyperparameters, we used `ray tune` along with `optuna`. The combination of these three already available solutions allowed us to perform multi-objective optimisation at scale, which was not possible in standalone `auto-sklearn`. Arguably, `autosklearn 2.0` fixes these issues, however, at the time of this writing, multi-objective optimisation is not an official feature in the package. Another choice is `FLAML`(Wang et al., 2021b), however, its documented moderate reliability (Gijsbers et al., 2024) finalised the choice of `auto-sklearn`. The OpenML CC-18 dataset and its features are fed into the Bayesian or random optimiser, or both (depending on the experiment conducted). These optimisers are provided by Optuna. Depending on the experiment, we also feed the random forest baseline metrics. Optuna generates hyperparameter configurations and Ray Tune runs them in parallel for each feature. An illustration of the architecture is given in Figure 10.

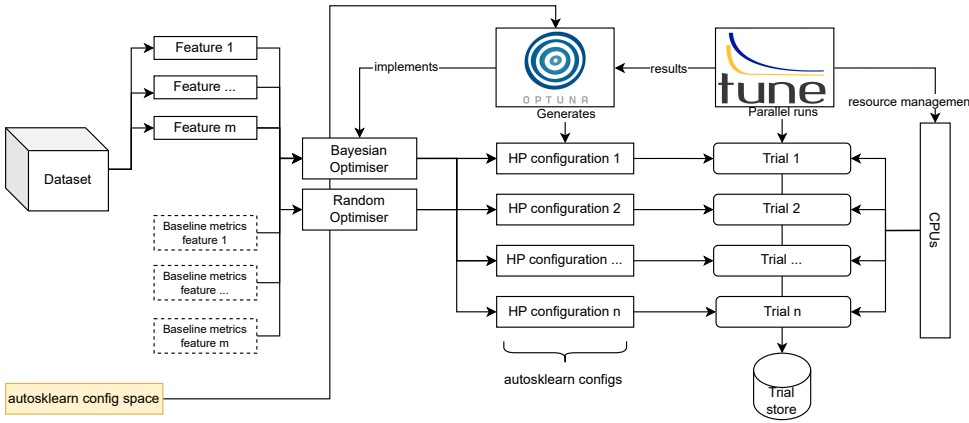

Figure 10: Architecture of our custom AutoML solution

**Extracting the search space**   To create our custom AutoML solution, we extracted the full search space of `autosklearn` library. The classification and pre-processing algorithms are given in Table 3 and Table 4, respectively. The library is built using `ConfigSpace` (Lindauer et al., 2019), which provides a simple function to extract the entire search space including classification algorithms, their hyperparameters, type of hyperparameter: categorical or continuous, and the conditions on them, if any.

**Optuna for multi-objective optimisation**   Optuna is an automatic hyperparameter optimization software framework, particularly designed for machine learning. It features an imperative, *define-by-run* style user API. Optuna features multi-objective optimisation and defines the following basic concepts.

- **Study**: optimization based on an objective function

- **Trial**: a single execution of the objective function

The goal of *study* is to find an optimal set of hyperparameter values through multiple *trials*.

We leverage the define-by-run style user API and recreate the extracted search space from `auto-sklearn`. This recreated search space can be used with `optuna` and the optimisation algorithms in the package, which support multiple objectives.

**Ray Tune for distributed computing of trials**   `Ray` (Moritz et al., 2018) is an open source framework to build and scale machine learning applications. `Tune` (also called `Ray Tune`), a sub-package of `Ray`, is a Python library for experimental

Table 3: Classification algorithms provided in `auto-sklearn`. The number of hyperparameters associated with the algorithm ($\lambda$) are separated into categorical (cat) and continuous (cont). The numbers in the parenthesis are the conditional hyperparameters, which are relevant when other parameters are active.

| Name | #$\lambda$ | cat (cond) | cont (cond) |
|---|---|---|---|
| AdaBoost (AB) | 4 | 1 (-) | 3 (-) |
| Bernoulli naïve Bayes | 2 | 1 (-) | 1 (-) |
| decision tree (DT) | 4 | 1 (-) | 3 (-) |
| extreml. rand. trees | 5 | 2 (-) | 3 (-) |
| Gaussian naïve Bayes | - | - | - |
| gradient boosting (GB) | 6 | - | 6 (-) |
| kNN | 3 | 2 (-) | 5 (-) |
| LDA | 4 | 1 (-) | 1 (-) |
| linear SVM | 4 | 2 (-) | 3 (-) |
| kernel SVM | 7 | 2 (-) | 5 (2) |
| multinomial naïve Bayes | 2 | 1 (-) | - |
| passive aggressive | 3 | 1 (-) | 2 (-) |
| QDA | 4 | 1 (-) | 1 (-) |
| random forest (RF) | 5 | 2 (-) | 3 (-) |
| Linear Class. (SGD) | 10 | 4 (-) | 6 (3) |

execution and hyperparameter tuning at any scale. We use `Ray Tune` to enable distributed execution of `Optuna` trials, saving trials, metadata and maintaining trial states during and after the end of a trial run.

## C. Weighted-sum scalarisation approach to multi-objective optimisation for X-hacking

The typical target of an automated machine learning pipeline is a model classification performance metric such as a confusion matrix, or derived statistics such as accuracy, $F_1$ score and area under the receiver operating characteristic (ROC) curve. At the same time, we wish to optimize our desired model explanation, which could be described through a $p$-value or coefficient value, feature importance, effect size (e.g. Cohen's $d$), Shapley value or fairness metric (Agrawal et al., 2021, § 4). The task is therefore a multi-objective optimization problem. How much predictive performance must be sacrificed, on average, to get the explanation desired? To explore the trade-off between accuracy (or AUC, $F_1$, Brier score etc.) against the chosen XAI metric (SHAP, $p$-value, Cohen's $\delta$), we propose a scalarized scoring function, $Q$, based on a weighted sum of *lying* (2), *performance* (3) and *obviousness* (4).

Let $\mathrm{perf}(m)$ denote the predictive performance of model pipeline $m$ and let $\mathrm{obv}(m)$ denote some quantified measure of obviousness, audacity or inadmissibility. Let $m_\uparrow := \arg\max_m\{\mathrm{perf}(m)\}$ be the best-performing pipeline and $m_\downarrow$ denote an acceptable baseline model[1]. Let $v$ denote the feature explained by an XAI metric $X_m(v)$, such as SHAP. Then we maximize the scalar objective function

$$Q := -\,\mathrm{sgn}\left(X_{m\uparrow}(v)\right) \cdot \lambda \cdot X_m(v) \tag{2}$$
$$+\,\mathrm{perf}(m) \tag{3}$$
$$-\,\mu \cdot \mathrm{obv}(m), \tag{4}$$

where $\lambda$ and $\mu$ are parameters that define a Pareto front of possible solutions. Such an objective may be passed to any AutoML framework that accepts custom scoring metrics.

As XAI and predictive performance metrics lie on different scales, efficient selection of $\lambda$ is crucial. Suppose that the 'best' model is anticipated to have accuracy no greater than 0.9, while a baseline model achieves accuracy of 0.7. If a baseline or prior SHAP is $+1$, and the aim is to find a model that 'flips' or nullifies this, then a pragmatic starting point is $\lambda \le (\mathrm{perf}(m_\uparrow) - \mathrm{perf}(m_\downarrow))/X_{m_\downarrow} = (0.9 - 0.7)/1 = 0.2$, so that larger-scale changes in SHAP 'drive' the optimizer.

---

[1] $m_\downarrow := \arg\min_m\{\mathrm{perf}(m)\}$ for acceptable models $m \in \mathcal{M}$

Table 4: Pre-processing algorithm provided in `auto-sklearn`. The number of hyperparameters associated with the algorithm ($\lambda$) are separated into categorical (cat) and continuous(cont). The numbers in the parenthesis are the conditional hyperparameters, which are relevant when other parameters are active.

| Name | #$\lambda$ | cat (cond) | cont (cond) |
|---|---|---|---|
| extreml. rand. trees prepr. | 5 | 2 (-) | 3 (-) |
| fast ICA | 4 | 3 (-) | 1 (1) |
| feature agglomeration | 4 | 3 (-) | 1 (-) |
| kernel PCA | 5 | 1 (-) | 4 (3) |
| rand. kitchen sinks | 2 | - | 4 (3) |
| linear SVM prepr. | 3 | 1 (-) | 2 (-) |
| no preprocessing | - | - | 4 |
| nystroem sampler | 5 | 1 (-) | 4 (3) |
| PCA | 2 | 1 (-) | 1 (-) |
| polynomial | 3 | 2 (-) | 1 (1) |
| random trees embed. | 4 | 2 (-) | 1 (-) |
| select percentile | 2 | 1 (-) | - |
| select rates | 3 | 2 (-) | - |
| one-hot encoding | 2 | 1 (-) | 1 (1) |
| imputation | 1 | 1 (-) | - |
| balancing | 1 | 1 (-) | - |
| rescaling | 1 | 1 (-) | - |

# D. Details on experimental setup and results

This section covers the implementation details and results from *post-hoc* and *ad-hoc* strategies for X-hacking. For each strategy used to perform X-hacking we first list the datasets used, give details on the resource and implementation and finally give the results for all the datasets used to perform X-hacking.

## D.1. Experimental setup and results for *post-hoc* X-hacking (Cherry-picking)

This section covers the implementation details and results from the experiments for all the datasets for *post-hoc* X-hacking. Following listing the datasets used and implementations details we show the graphs for changes in feature importance relative to baseline for all the datasets, lines of best fit of SHAP for all the features for all the datasets, and marginal distribution of SHAP slopes over AutoML-evaluated models for all the features of all the datasets.

### D.1.1. DATASETS

For the current experiment, we have 23 datasets from OpenML Benchmark CC-18 (Bischl et al., 2021). The current experiment for explainability of the algorithms are carried out on datasets with a binary target. The datasets are listed in Table 5.

### D.1.2. RESOURCE AND IMPLEMENTATION DETAILS

For all of the calculations, we stick to parallel computation using CPUs. For our experiments we have used at most 192 CPUs in parallel on an institutional computing cluster. Since the datasets are structured, a maximum of 300 GB of RAM was used for the experiments. Since computation time for SHAP calculations increase as one increases the number of test samples, parallelism become imperative. We restricted the number of test samples to 100 to calculate SHAP for all the datasets for resource management reasons.

The implementation is done in `Python` programming language. We used `pandas` and `numpy` libraries for data wrangling, `scikit-learn` as our base ML library, `auto-sklearn` for automated finding of models, `optuna` for multi-objective optimisation, and `shap` for calculating shap values.

Table 5: Binary classification datasets from OpenML CC-18 Benchmark suite.

| OpenML ID | Dataset | #Features |
|---:|---|---:|
| 15 | breast-w | 9 |
| 29 | credit-approval | 15 |
| 31 | credit-g | 20 |
| 37 | diabetes | 8 |
| 1049 | pc4 | 37 |
| 1050 | pc3 | 37 |
| 1053 | jm1 | 21 |
| 1063 | kc2 | 21 |
| 1067 | kc1 | 21 |
| 1068 | pc1 | 21 |
| 1461 | bank-marketing | 16 |
| 1464 | blood-transfusion | 4 |
| 1480 | ilpd | 10 |
| 1485 | madelon | 500 |
| 1494 | qsar-biodeg | 41 |
| 1510 | wdbc | 30 |
| 1590 | adult | 14 |
| 4134 | Bioresponse | 1776 |
| 23517 | numerai28.6 | 21 |
| 40701 | churn | 20 |
| 40983 | wilt | 5 |
| 40994 | climate-model | 18 |
| 45547 | cardiac-disease | 12 |

**data split** : for training all the models, baseline and AutoML, we used 20% of the samples as test dataset for all of the datasets mentioned in Table 5.

**preprocessing** : for all of the datasets, the samples where any feature had missing (NaN) values were removed. The indices of the omitted data are saved for later results.

**random seed** : for reproducibility of the results, a random seed of 42 is used everywhere.

**baseline** : the baseline model is the default `sklearn.ensemble.RandomForestClassifier` with the mentioned random state.

**AutoML** : for each dataset, we run `auto-sklearn` for 3600 seconds in total and a run time limit of 100 seconds for each candidate model with the mentioned random seed. An ensemble size of 1 is used since ensemble models are out of the scope for the current experiment. A model that takes more than 100 seconds to train is omitted.

**explainer** : due to its model agnostic behaviour, we use `shap.KernelExplainer` for calculating the SHAP values. A background sample of 50 and test sample of 100 samples from the test split is used. The respective indices of the 100 samples are saved for regression analysis discussed later.

### D.1.3. RESULTS

In this sub-section we show all the graphs for the experiment conducted on all of the datasets. Due to large number of features in the dataset `Bioresponse` we do not include the graphs for them here. However, they are available in the GitHub repository. Similarly, since there are too many visualizations for lines of best fits and kernel density graphs for fitted slopes for SHAP values, we include a few examples here and the rest are available on GitHub.

**Changes in feature importance relative to baseline** The changes in feature importance relative to the baseline for a feature $v$ for an AutoML model is calculated as:

$$y = \frac{\frac{|\ \overline{\text{SHAP}}_v\ |_{\text{baseline}}}{|V|}}{\sum_{j=1}^{|V|} |\ \overline{\text{SHAP}}_j\ |_{\text{baseline}}} - \frac{\frac{|\ \overline{\text{SHAP}}_v\ |_{\text{automl}}}{|V|}}{\sum_{j=1}^{|V|} |\ \overline{\text{SHAP}}_j\ |_{\text{automl}}} \tag{5}$$

where $V$ is the set of all features for a dataset. This difference gives us the relative change in explainability from the baseline model for each feature in all the datasets. This can be visualised through violin plots for all the datasets. The violin plots are given below.

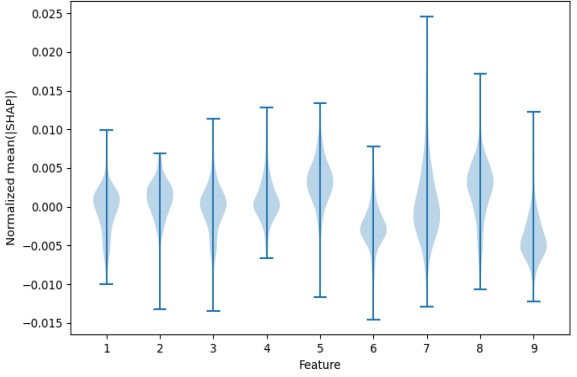

Figure 11: Relative change for dataset `breast-w`

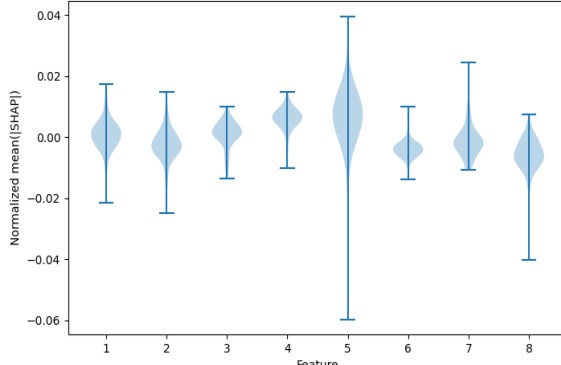

Figure 12: Relative change for dataset `credit-approval`

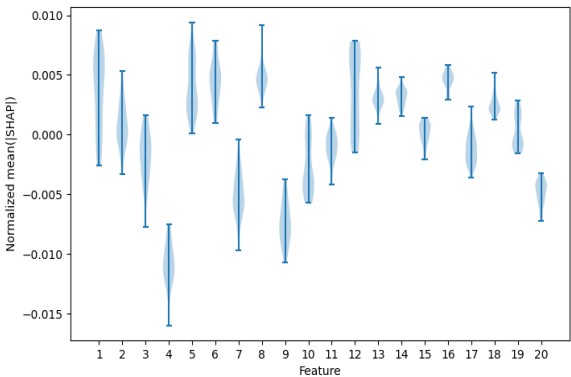
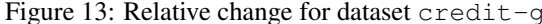

Figure 13: Relative change for dataset `credit-g`

Figure 14: Relative change for dataset `diabetes`

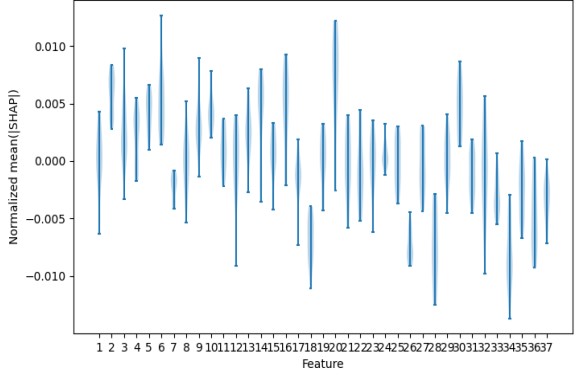

Figure 15: Relative change for dataset `pc4`

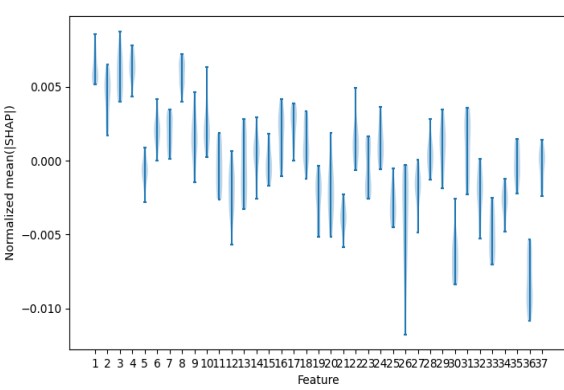

Figure 16: Relative change for dataset `pc3`

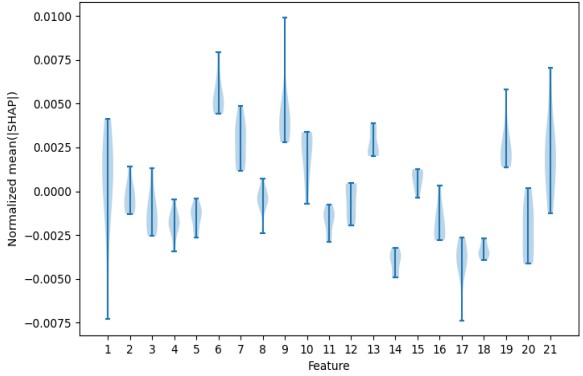

Figure 17: Relative change for dataset `jm1`

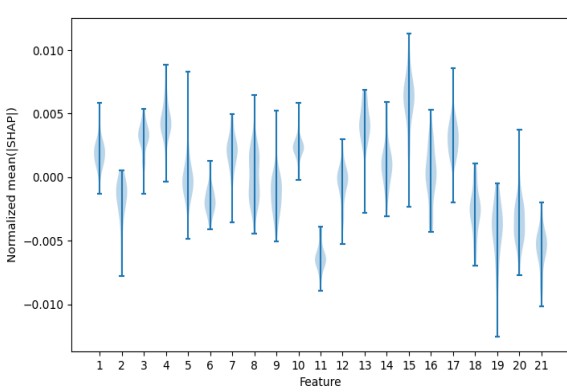

Figure 18: Relative change for dataset `kc2`

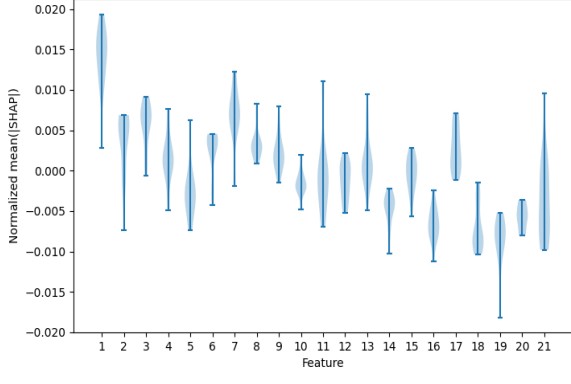

Figure 19: Relative change for dataset `kc1`

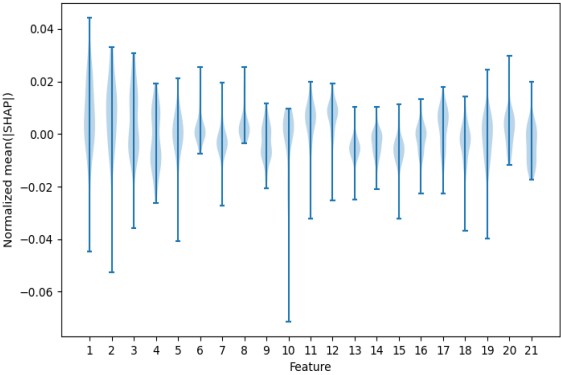

Figure 20: Relative change for dataset `pc1`

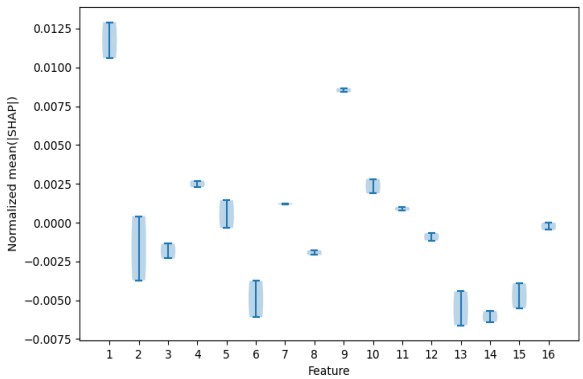

Figure 21: Relative change for dataset `bank-marketing`

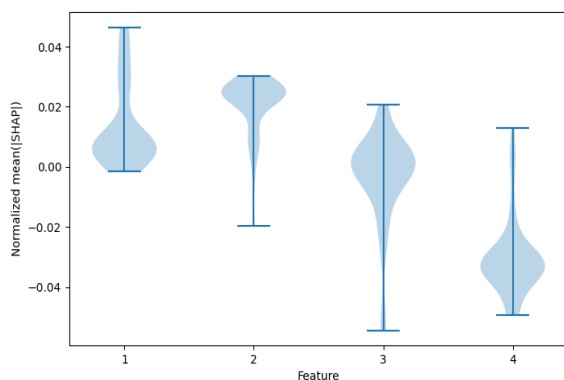

Figure 22: Relative change for dataset `blood-transfusion`

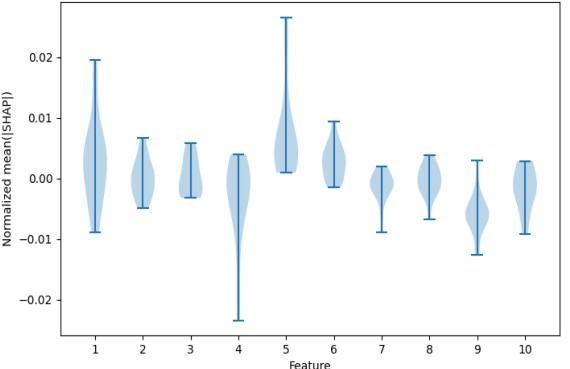

Figure 23: Relative change for dataset `ilpd`

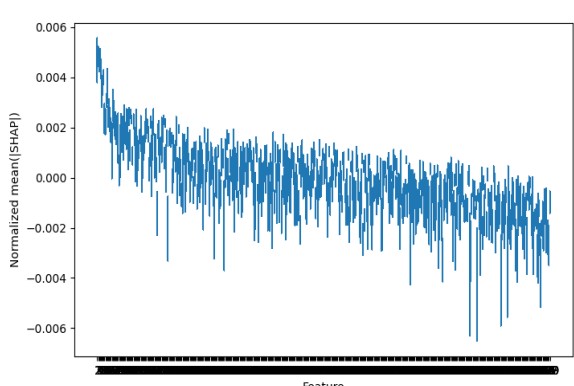

Figure 24: Relative change for dataset `madelon`

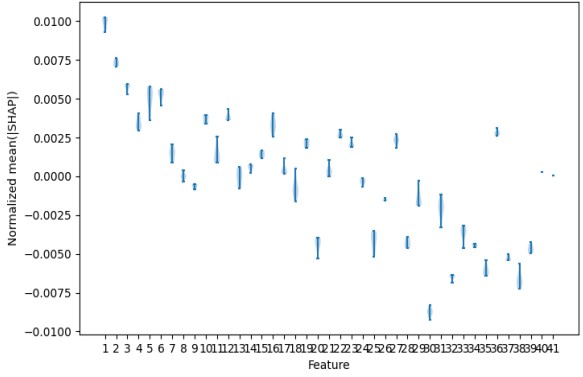
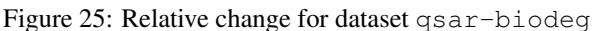

Figure 25: Relative change for dataset `qsar-biodeg`

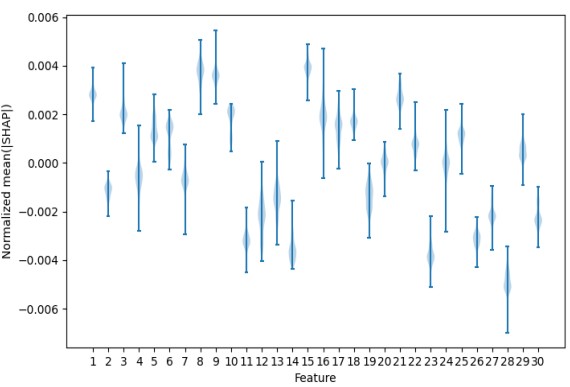

Figure 26: Relative change for dataset `wdbc`

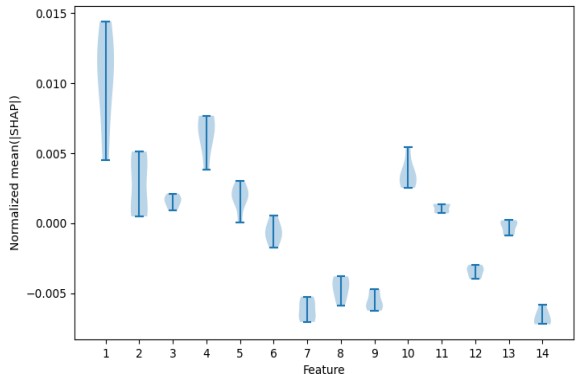

Figure 27: Relative change for dataset `adult`

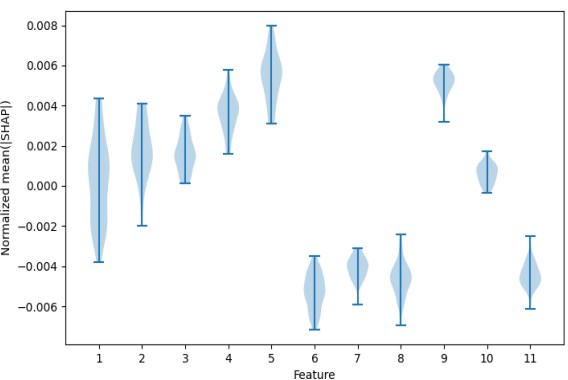

Figure 28: Relative change for dataset `cardiac-disease`

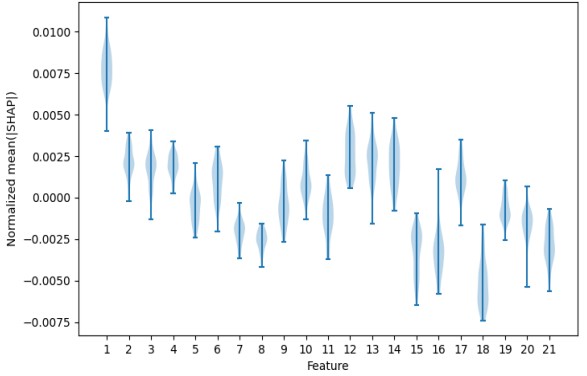

Figure 29: Relative change for dataset `numerai28.6`

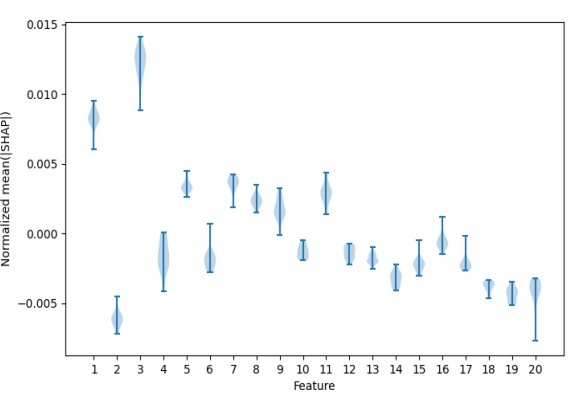

Figure 30: Relative change for dataset `churn`

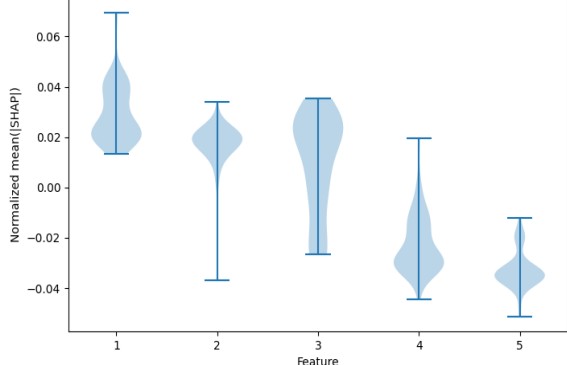

Figure 31: Relative change for dataset `wilt`

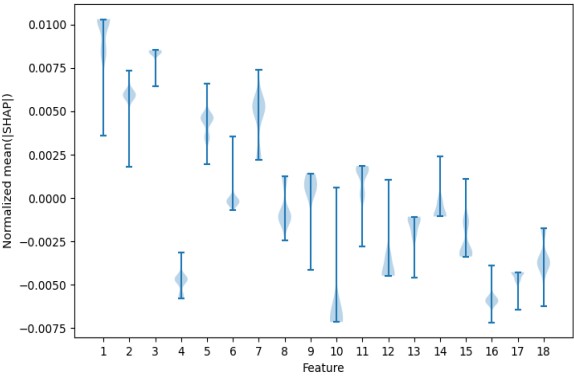

Figure 32: Relative change for dataset `climate-model`

**Lines of best fit of SHAP**    To calculate the lines of best fit for SHAP for all features of a dataset, the subset of the test split on which the SHAP values were calculated is taken as the data with the corresponding SHAP values as the target. A linear regression model is fitted on this data using `sklearn.linear_model.LinearRegression` for the baseline as well as all the models that have accuracy at least as good as baseline found by running AutoML on the dataset. Since displaying all the graphs is not feasible, we provide with a few examples here. The lines of best fit for the rest of the features for all of the datasets can be seen in the GitHub repository.

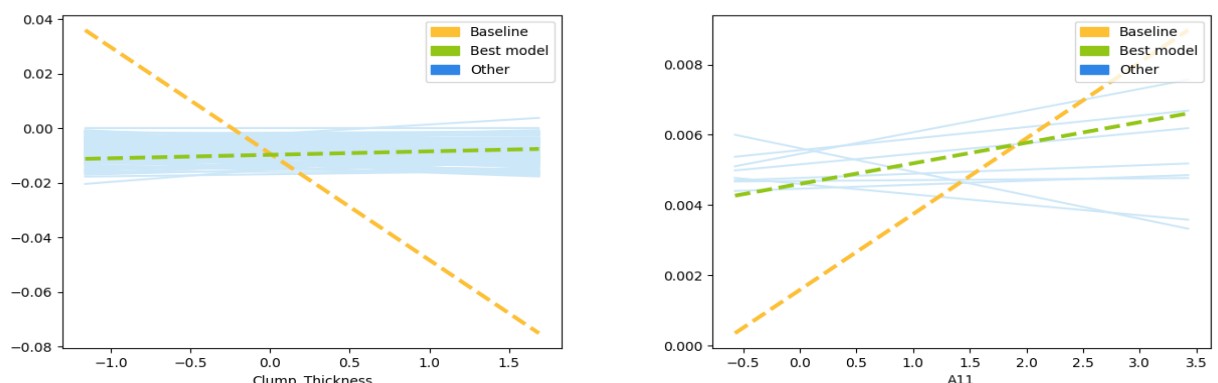

Figure 33: Lines of best fit of SHAP for feature `Clump Thickness` in dataset `breast-w`

Figure 34: Lines of best fit of SHAP for feature `A11` in dataset `credit-approval`

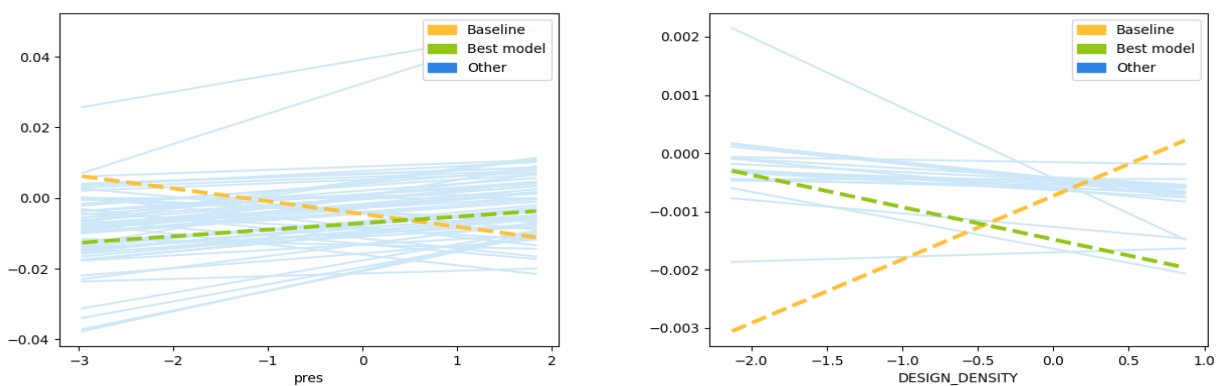

Figure 35: Lines of best fit of SHAP for feature `pres` in dataset `diabetes`

Figure 36: Lines of best fit of SHAP for feature `Design Density` in dataset `pc4`

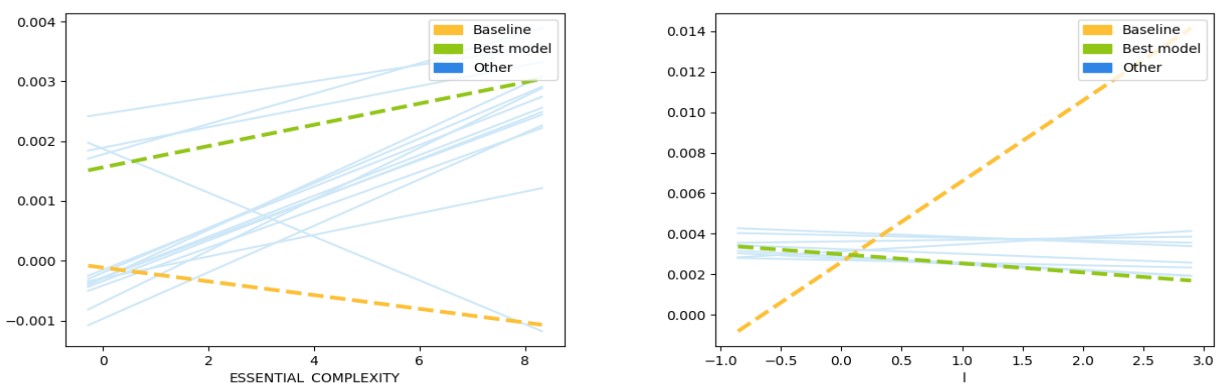

Figure 37: Lines of best fit of SHAP for feature `Essential Complexity` in dataset `pc3`

Figure 38: Lines of best fit of SHAP for feature `l` in dataset `jm1`

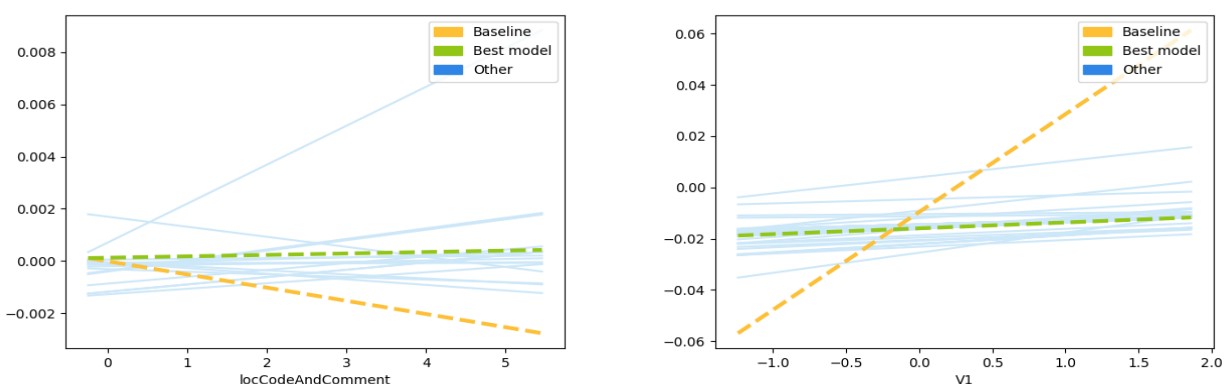

Figure 39: Lines of best fit of SHAP for feature `locCodeAndComment` in dataset `kc1`

Figure 40: Lines of best fit of SHAP for feature `V1` in dataset `blood-transfusion`

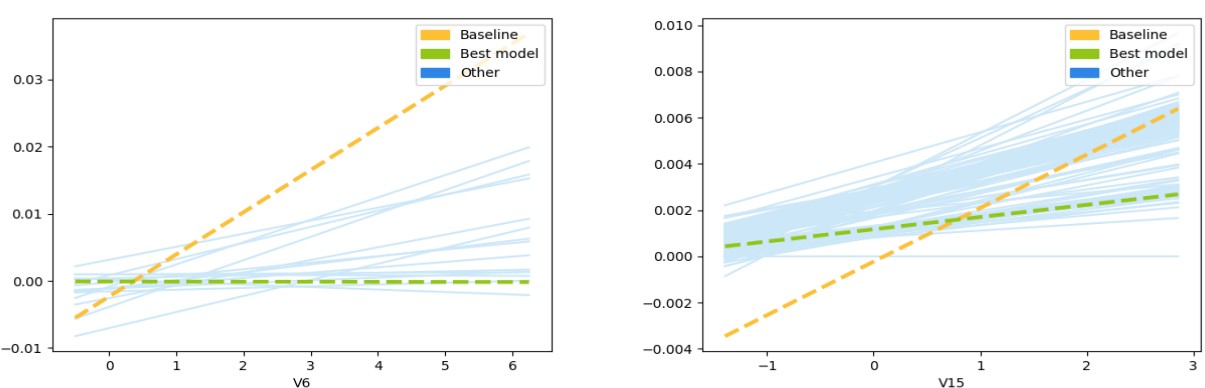

Figure 41: Lines of best fit of SHAP for feature `V6` in dataset `ilpd`

Figure 42: Lines of best fit of SHAP for feature `V15` in dataset `wdbc`

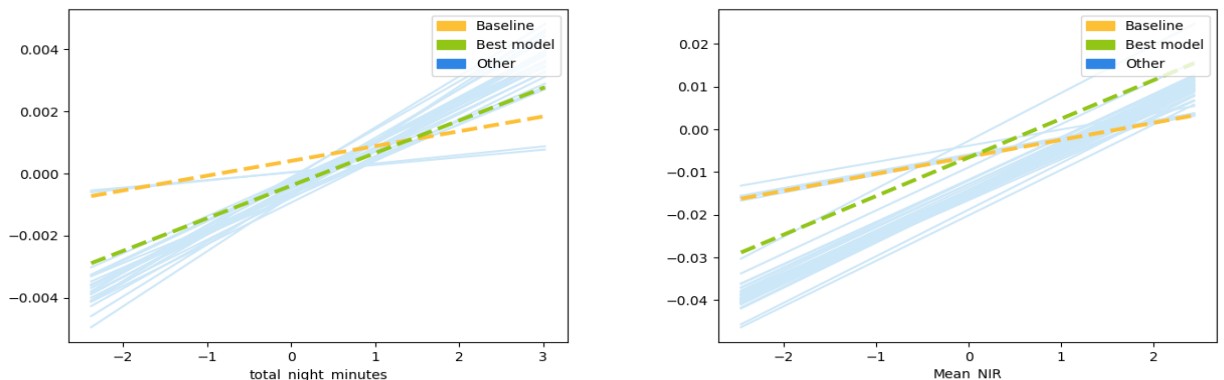

Figure 43: Lines of best fit of SHAP for feature total_night_minutes in dataset churn

Figure 44: Lines of best fit of SHAP for feature Mean_NIR in dataset wilt

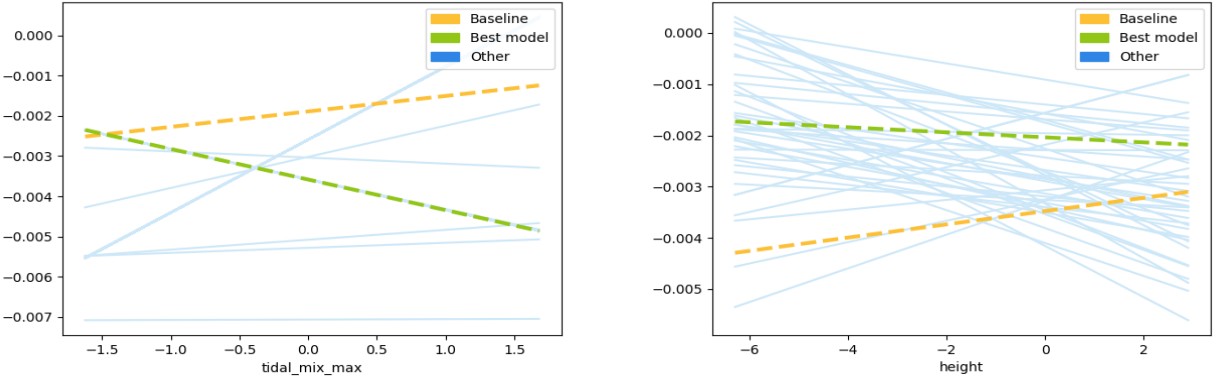

Figure 45: Lines of best fit of SHAP for feature tidal_mix_max in dataset climate-model

Figure 46: Lines of best fit of SHAP for feature height in dataset cardiac-disease

Here we show the marginal distribution of slopes over all Auto-ML models that have the accuracy at least as good as the baseline model. The graphs correspond to the ones displayed in the previous sub-section.

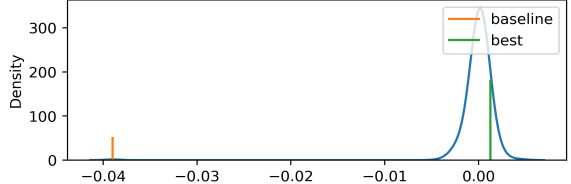
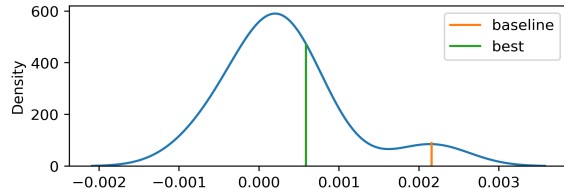

Figure 47: Marginal distribution of SHAP slopes for feature `Clump_Thickness` in dataset `breast-w`

Figure 48: Marginal distribution of SHAP slopes for feature `A11` in dataset `credit-approval`

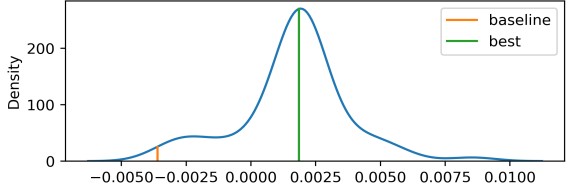
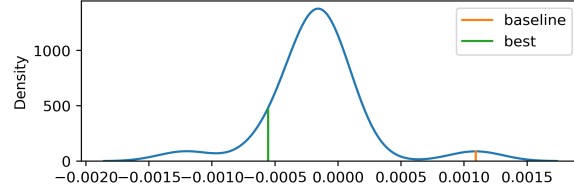

Figure 49: Marginal distribution of SHAP slopes for feature `pres` in dataset `diabetes`

Figure 50: Marginal distribution of SHAP slopes for feature `Design Density` in dataset `pc4`

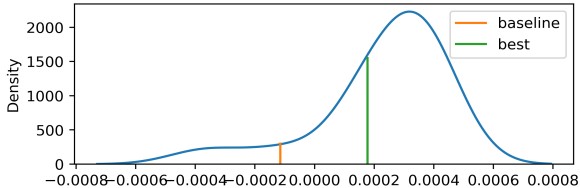
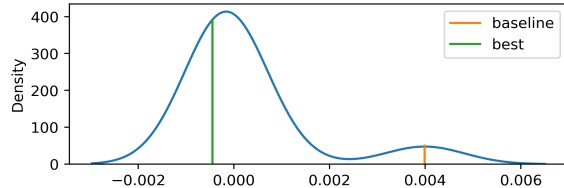

Figure 51: Marginal distribution of SHAP slopes for feature `Essential Complexity` in dataset `pc3`

Figure 52: Marginal distribution of SHAP slopes for feature `l` in dataset `jm1`

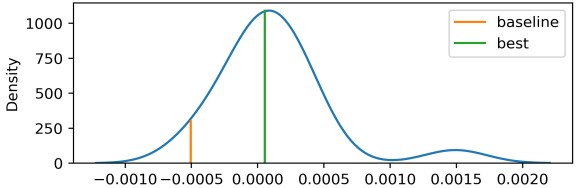

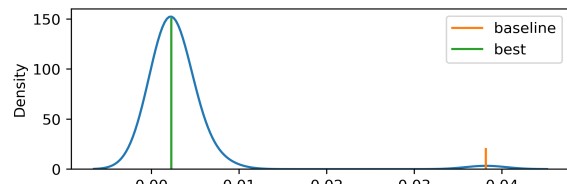

Figure 53: Marginal distribution of SHAP slopes for feature `locCodeAndComment` in dataset `kc1`

Figure 54: Marginal distribution of SHAP slopes for feature `V1` in dataset `blood-transfusion`

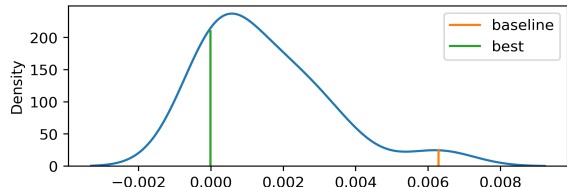

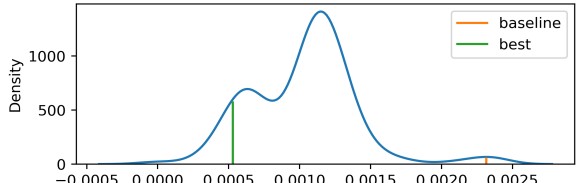

Figure 55: Marginal distribution of SHAP slopes for feature `V6` in dataset `ilpd`

Figure 56: Marginal distribution of SHAP slopes for feature `V15` in dataset `wdbc`

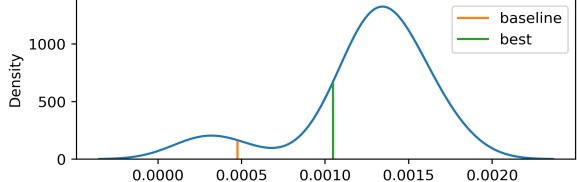

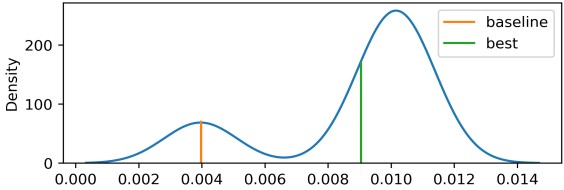

Figure 57: Marginal distribution of SHAP slopes for feature `total_night_minutes` in dataset `churn`

Figure 58: Marginal distribution of SHAP slopes for feature `Mean_NIR` in dataset `wilt`

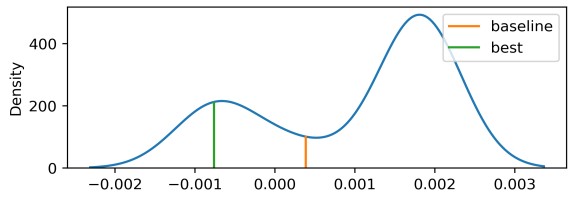

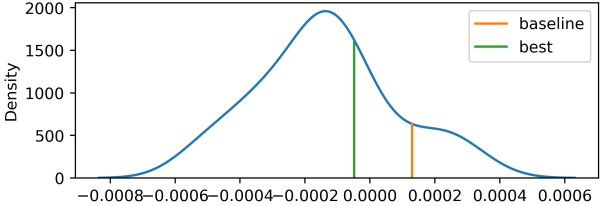

Figure 59: Marginal distribution of SHAP slopes for feature `tidal_mix_max` in dataset `climate-model`

Figure 60: Marginal distribution of SHAP slopes for feature `height` in dataset `cardiac-disease`

## D.2. Experimental setup and results for *ad-hoc* X-hacking (Directed search)

This section covers the implementation details and results from the experiments for all the datasets for *ad-hoc* X-hacking. Following listing the datasets used and implementations details, we show the graphs for cumulative minimum of mean absolute SHAP for Bayesian optimisation and random sampling. These graphs are shown for top-4 features from each dataset according to the random forest baseline.

## D.3. Dataset

The datasets are the same as given in § D.1.1

## D.4. Resource and implementation details

For all of the calculations, we stick to parallel computation using CPUs. For *ad-hoc* X-hacking we have used at most 292 CPUs in parallel on an institutional computing cluster. To run many candidate models for each dataset, a maximum of 1 TB of RAM was used for the experiments. Since computation time for SHAP calculations increase as one increases the number of test samples, parallelism become imperative. We restricted the number of test samples to 100 to calculate SHAP for all the datasets, for resource management reasons.

The implementation is done in `Python` programming language. We used `pandas` and `numpy` libraries for data wrangling, `scikit-learn` as our base ML library, `auto-sklearn` for its search space including models and its hyperparameters, `ConfigSpace` to extract the search space from `auto-sklearn`, `optuna` for enabling multi-objective optimisation, `ray tune` for parallel running of different models and intermediate storage of experiment related metrics, and `shap` for calculating shap values.

**data split** : for training all the models, baseline and models from our custom AutoML implementation, we used 20% of the samples as test dataset for all the datasets mentioned in Table 5.

**preprocessing** : for all the datasets, the samples where any feature had missing (NaN) values were removed. The indices of the omitted data are saved for later results.

**random seed** : for reproducibility of the results, a random seed of 42 is used everywhere.

**baseline** : the baseline model is the default `sklearn.ensemble.RandomForestClassifier` with the mentioned random state.

**AutoML** : for each dataset, we run our custom AutoML solution described in § B for 12 hours (43200 seconds) in total and a run time limit of 1 hour (3600 seconds) for each candidate model with the mentioned random seed. We the custom AutoML solution with both Bayesian optimisation and random sampling

**explainer** : due to its model agnostic behaviour, we use `shap.KernelExplainer` for calculating the SHAP values. A background sample of 50 and test sample of 100 samples from the test split is used. The respective indices of the 100 samples are saved for regression analysis discussed later.

### D.4.1. RESULTS

In this sub-section we show using graphs for cumulative minimum of mean absolute SHAP for Bayesian optimisation and random sampling for all the datasets. It is expected that some features are not vulnerable to X-hacking, and thus for those features we do not see the data in the plots[2].

---

[2]empty graphs represent features being robust to X-hacking. Marked with *

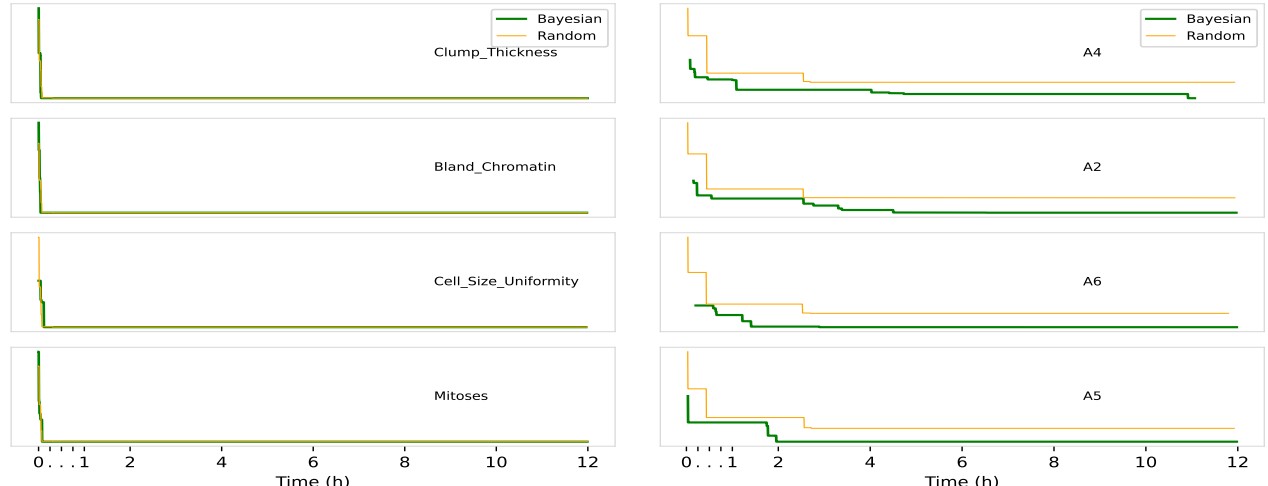

Figure 61: Cumulative minimum of mean absolute SHAP for top 4 features of dataset `breast-w` for Bayesian optimisation and random sampling

Figure 62: Cumulative minimum of mean absolute SHAP for top 4 features of dataset `credit-approval` for Bayesian optimisation and random sampling

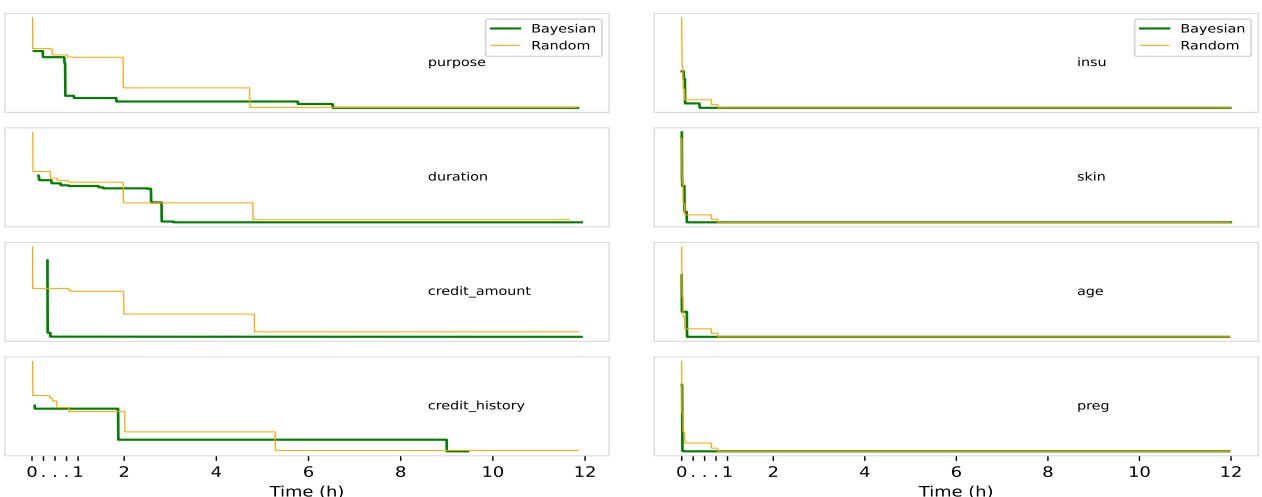

Figure 63: Cumulative minimum of mean absolute SHAP for top 4 features of dataset `credit-g` for Bayesian optimisation and random sampling

Figure 64: Cumulative minimum of mean absolute SHAP for top 4 features of dataset `diabetes` for Bayesian optimisation and random sampling

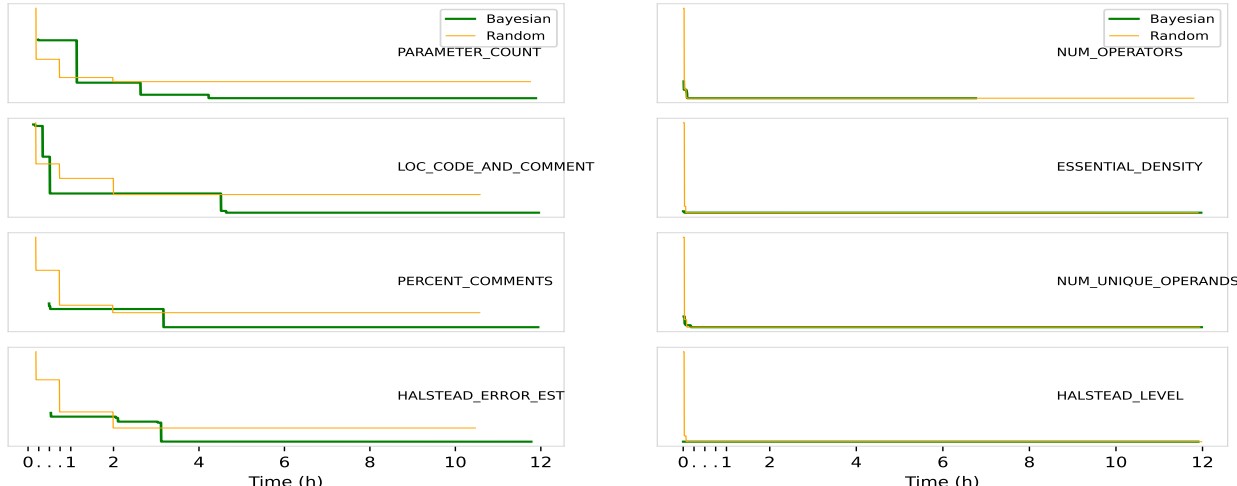

Figure 65: Cumulative minimum of mean absolute SHAP for top 4 features of dataset `pc4` for Bayesian optimisation and random sampling

Figure 66: Cumulative minimum of mean absolute SHAP for top 4 features of dataset `pc3` for Bayesian optimisation and random sampling

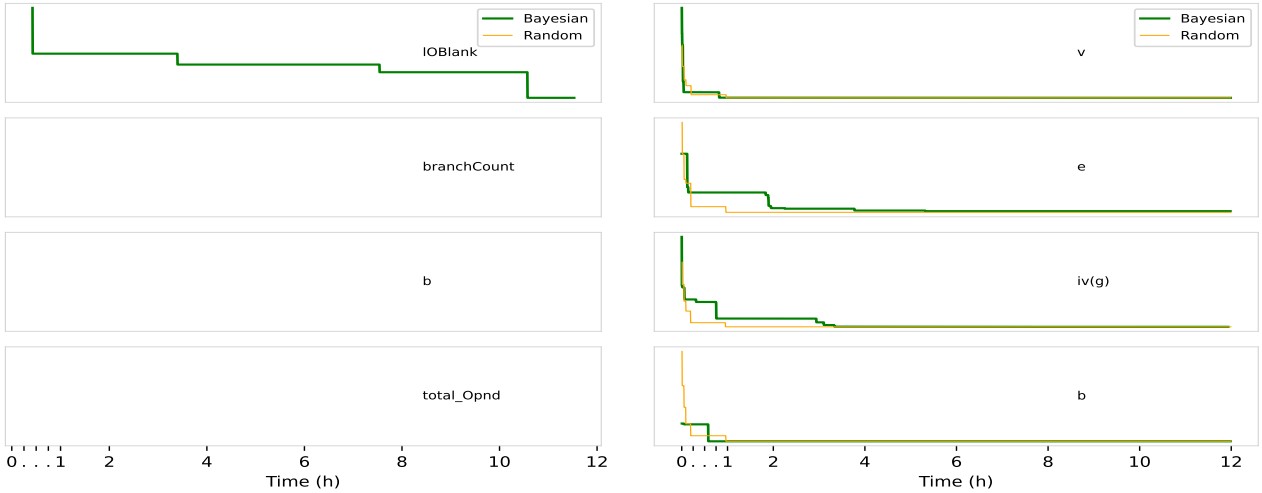

Figure 67: Cumulative minimum of mean absolute SHAP for top 4 features of dataset `jm1` for Bayesian optimisation and random sampling*

Figure 68: Cumulative minimum of mean absolute SHAP for top 4 features of dataset `kc2` for Bayesian optimisation and random sampling

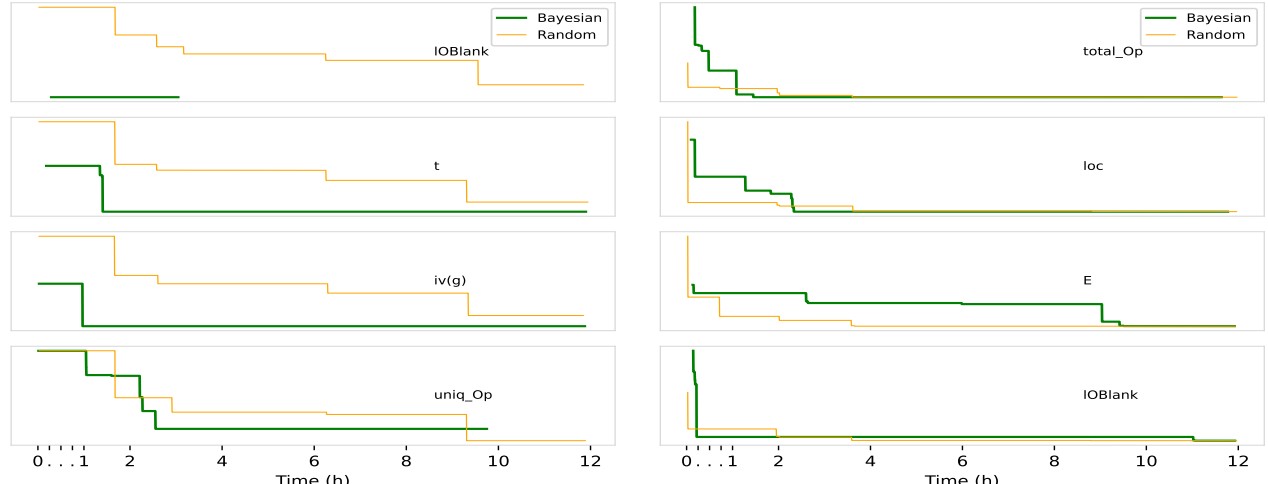

Figure 69: Cumulative minimum of mean absolute SHAP for top 4 features of dataset `kc1` for Bayesian optimisation and random sampling

Figure 70: Cumulative minimum of mean absolute SHAP for top 4 features of dataset `pc1` for Bayesian optimisation and random sampling

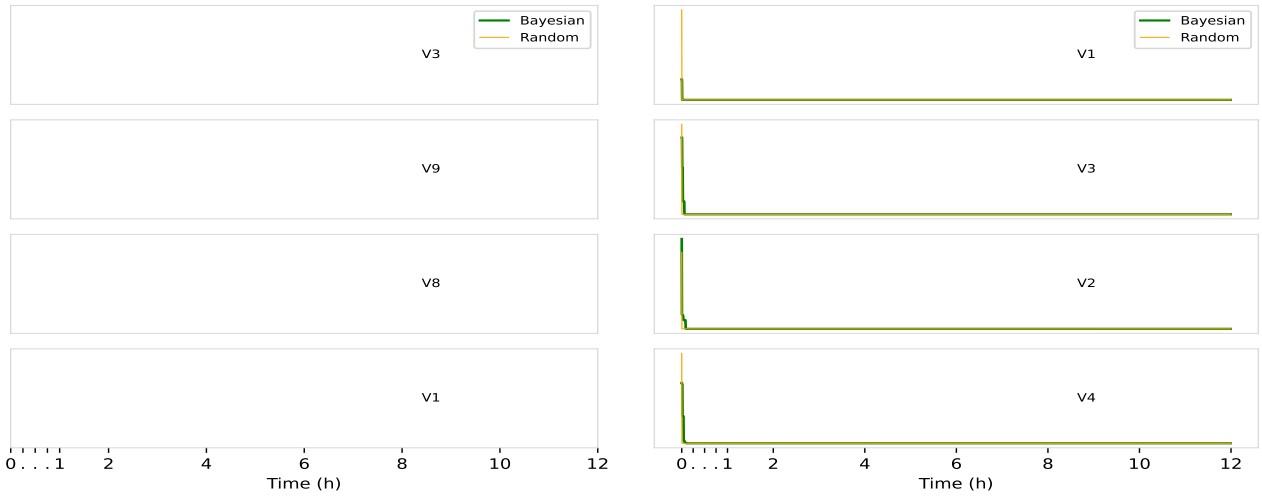

Figure 71: Cumulative minimum of mean absolute SHAP for top 4 features of dataset `bank-marketing` for Bayesian optimisation and random sampling*

Figure 72: Cumulative minimum of mean absolute SHAP for top 4 features of dataset `blood-transfusion` for Bayesian optimisation and random sampling

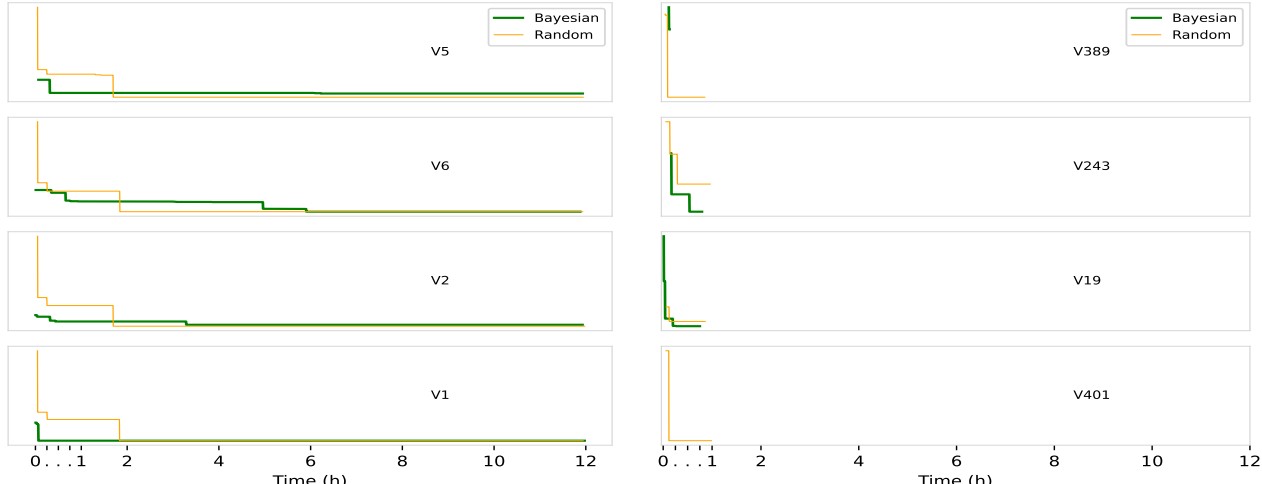

Figure 73: Cumulative minimum of mean absolute SHAP for top 4 features of dataset `ilpd` for Bayesian optimisation random sampling

Figure 74: Cumulative minimum of mean absolute SHAP for top 4 features of dataset `madelon` for Bayesian optimisation and random sampling

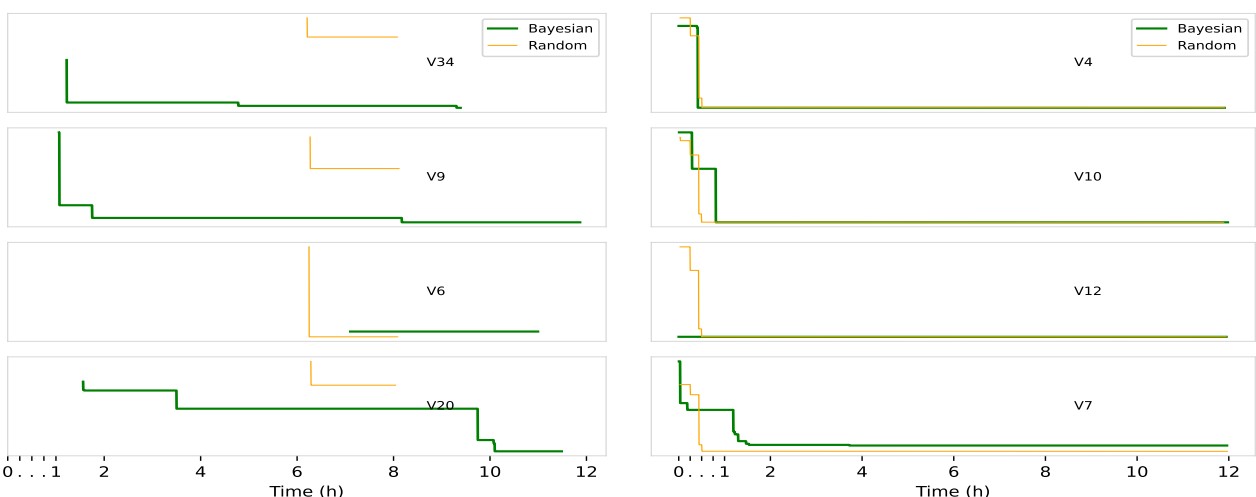

Figure 75: Cumulative minimum of mean absolute SHAP for top 4 features of dataset `qsar-biodeg` for Bayesian optimisation and random sampling

Figure 76: Cumulative minimum of mean absolute SHAP for top 4 features of dataset `wdbc` for Bayesian optimisation and random sampling

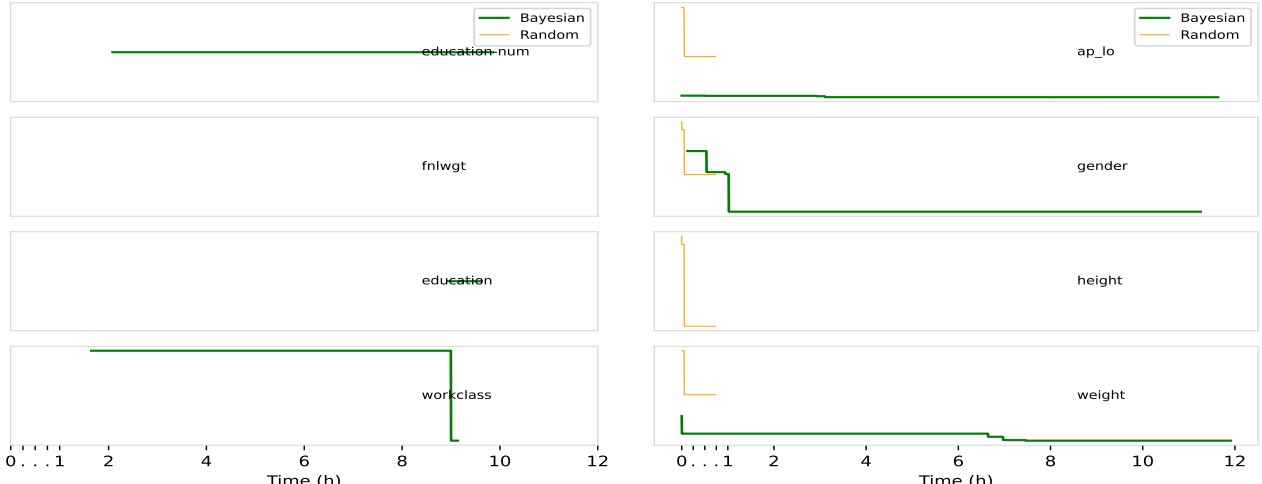

Figure 77: Cumulative minimum of mean absolute SHAP for top 4 features of dataset `adult` for Bayesian optimisation and random sampling*

Figure 78: Cumulative minimum of mean absolute SHAP for top 4 features of dataset `cardiac-disease` for Bayesian optimisation and random sampling

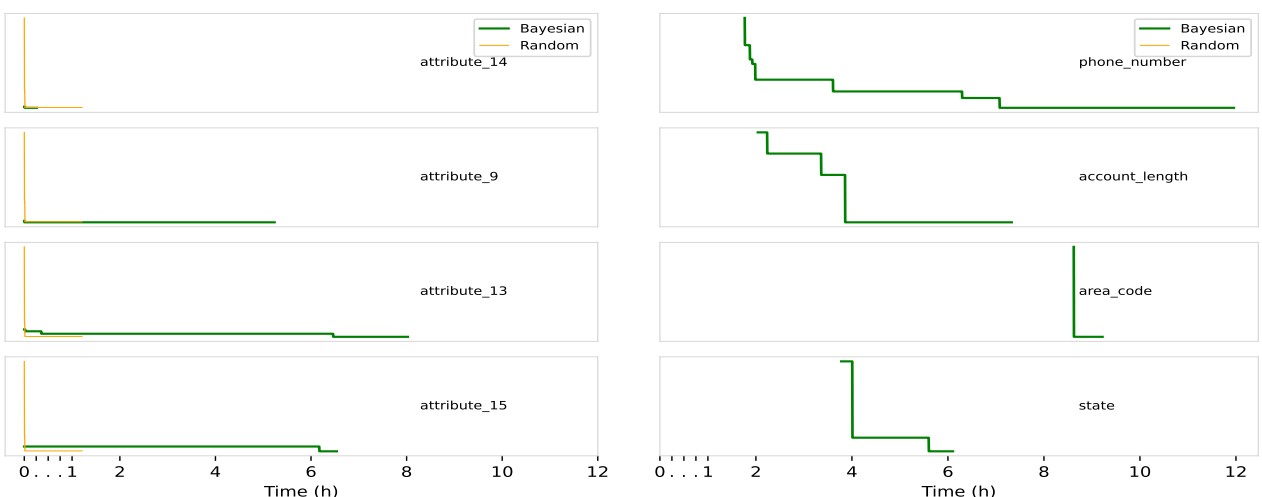

Figure 79: Cumulative minimum of mean absolute SHAP for top 4 features of dataset `numerai28.6` for Bayesian optimisation and random sampling

Figure 80: Cumulative minimum of mean absolute SHAP for top 4 features of dataset `churn` for Bayesian optimisation and random sampling

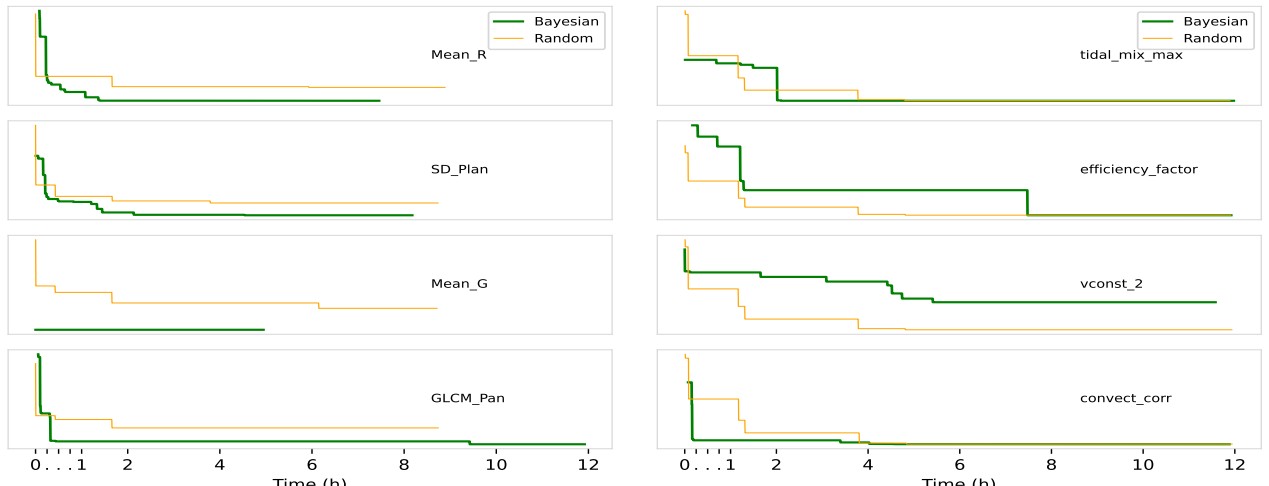

Figure 81: Cumulative minimum of mean absolute SHAP for top 4 features of dataset `wilt` for Bayesian optimisation and random sampling

Figure 82: Cumulative minimum of mean absolute SHAP for top 4 features of dataset `climate-model` for Bayesian optimisation and random sampling

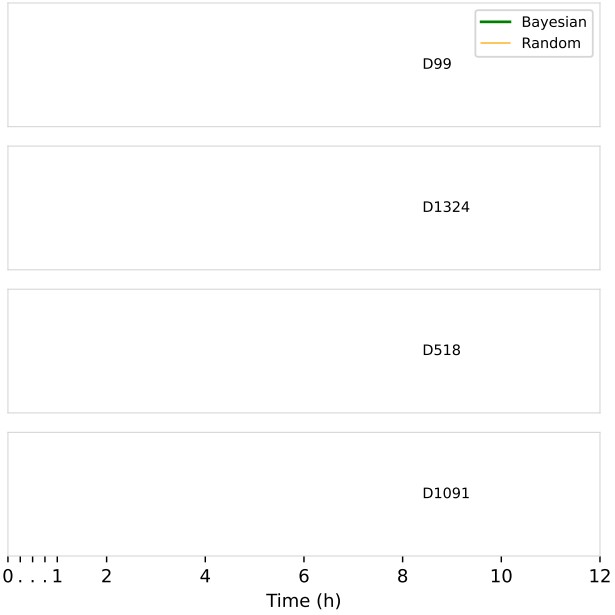

Figure 83: Cumulative minimum of mean absolute SHAP for top 4 features of dataset `Bioresponse` for Bayesian optimisation and random sampling*

## E. Exploring Shap Value Sensitivity to Noise and Information Sharing

In the first scenario, we examine a case featuring two identical features with no inherent independent noise. In this context, we observe no discernible shift in accuracy across different variables. This outcome can be attributed to the intrinsic additive nature of SHAP values, whereby the transfer of SHAP values from one feature to another occurs seamlessly without any resultant loss of accuracy. This is principally due to the absence of information loss.

The second experimental model introduces a lower noise-to-signal ratio, distinguishing between noise (independent information associated solely with a specific feature) and signal (information shared between both variables). When we transfer SHAP values from one variable to another within this context, we encounter a scenario where the transference fails to capture identical information. Consequently, accuracy experiences a decline, albeit not of a significant magnitude.

In the third experiment, we explore a high noise-to-signal ratio, leading to a noteworthy decrease in accuracy when manipulating SHAP values across different features. This emphasizes the sensitivity of SHAP values to the presence of substantial noise within the data.

These experiments are strategically designed to demonstrate the circumstances under which it is feasible to manipulate SHAP values for a given dataset without incurring substantial accuracy loss. It becomes easier to reduce or alter SHAP values for a specific feature when the information associated with that feature is shared among other variables.

In the context of an automated machine learning (AutoML) pipeline, different preprocessing steps and algorithms may yield models with comparable accuracy but varying feature importance, as indicated by SHAP values. The likelihood of encountering such models is contingent upon the search space of available models and the non-linearity inherent in the relationships among features or the underlying data generation process. For instance, when the relationship is profoundly non-linear, and the model's search space predominantly consists of linear models, these models are more prone to assigning higher SHAP importance to linear relationships within the data generation process.

the equations are

$$
\begin{aligned}
f_0 &\sim \text{Uniform}(0, 5) \\
f_1 &\sim 10 \cdot f_0 + \mathcal{N}(0, \sigma_1) \\
f_2 &\sim 20 \cdot f_0 + \mathcal{N}(0, \sigma_2) \\
f_3 &\sim 3 \cdot f_1 + 4 \cdot f_2 + \mathcal{N}(0, 0.01)
\end{aligned}
\tag{6}
$$

The variation in the normal distribution's standard deviation can be harnessed as a mechanism for adjusting the level of independent information within dependent variables. By employing this approach, we can simulate diverse scenarios representing distinct degrees of independent information.

Our empirical findings clearly illustrate that as the magnitude of independent information in the dependent variables is augmented, the corresponding reduction in predictive accuracy resulting from changes in SHAP (SHapley Additive exPlanations) values also experiences a corresponding increase. This relationship serves as compelling evidence of the critical role played by independent information in shaping the accuracy of predictive models, particularly in the context of SHAP value manipulation.

## F. Leveraging Redundancy for Shapley Value Manipulation

Incorporating a surplus of redundant variables into a model can be a strategically advantageous approach, particularly when the objective is to reverse the prevailing trend exhibited by Shapley values associated with a specific feature. This concept hinges on the premise that each child variable linked to the feature of interest encapsulates not only the information inherent to that feature but also additional information contributed by the respective child variable. For the sake of simplification and rigorous analysis, one can consider the information stemming from child nodes as a form of 'noise', while the data originating directly from the feature of interest, acting as the 'parent' variable, is regarded as the 'signal'. The overarching goal is to amplify the signal-to-noise ratio for the feature by strategically configuring different combinations of child variables.

In the ensuing illustrative examples, we expound upon this concept by showcasing the characteristic trends observed in Shapley values for distinct input values of specific features, achieved by fitting approximate linear regression models. A notable scenario arises when one child variable introduces an additional layer of information that follows a disparate trend

compared to another child variable linked to the same parent. In such instances, a linear amalgamation of these variables serves the purpose of disentangling the signal from the noise. This amalgamation strategically leverages the opposing noise effects exhibited in different models, thereby neutralizing their individual impact. If the noise effects happen to be multiplicative in nature, log transforms may be judiciously employed before engaging in linear combinations to attain similar results.

A foundational principle emerges from this exploration: the greater the number of independent child nodes associated with a specific feature, the more susceptible that feature becomes to a phenomenon referred to as "flipping", where its trend contrasts with the true underlying trend. Redundancy in the information domain empowers us to reconstruct the feature and construct a model that effectively showcases an inverse trend for the given feature values. Notably, this transformation can be accomplished while maintaining an equivalent level of predictive accuracy, as exemplified in the earlier instances illustrating the Shapley value versus accuracy Pareto front.

