# OpenReview forum: "X-Hacking: The Threat of Misguided AutoML"
_ICML.cc/2025/Conference — ICML 2025 poster_

### Official Review · Reviewer_vv4C · 2025-03-11

**Overall Recommendation:** 5

**Summary:**

The paper introduces the concept of X-Hacking, which refers to the practice of deliberately exploiting model multiplicity, where different ML models can have comparable performance, to select a model that has certain desirable explainability characteristics (like SHAP). The paper demonstrates the ability to select models based on explainable AI (XAI), discusses when this ability to select based on XAI is more feasible, and suggests some ways to make this selection more transparent.

**Claims And Evidence:**

The authors convincingly demonstrate that varying SHAP explanations can be obtained through either “cherry-picking” from multiple similarly performing models or via a targeted optimization approach (“directed-search”). They effectively show that this flexibility arises due to correlated or redundant features.

However, the paper's analogy to p-hacking raises conceptual difficulties. Unlike p-values, which are fundamentally about statistical inference, SHAP explanations are inherently model-specific reflections of what each model genuinely uses in prediction. Selecting a model because it uses some characteristics rather than others (as accurately represented by SHAP) is not inherently problematic. For instance, if policy explicitly mandates avoiding certain protected characteristics (like criminal record), choosing an alternative model that does precisely that is compliance—not manipulation. SHAP explanations truthfully reflect model behavior and do not inherently mislead; thus, selecting among models based on desirable explanations isn't automatically problematic.

A useful analogy is found in recent fairness literature, where “model multiplicity” is positively leveraged to select models with similar predictive power but improved fairness metrics. There, selecting among multiple models is explicitly beneficial, as it enables analysts to choose ethically or legally preferable options. This illustrates clearly that model selection itself—even based on explanation—can legitimately fulfill regulatory or ethical goals, provided it substantively addresses regulatory concerns.

Thus, the concern raised by the authors about “X-hacking” needs careful delineation: it is not simply choosing among multiple valid models that is problematic, but specifically doing so to create a misleading impression of objectivity or robustness when reporting explanations. This distinction warrants greater emphasis and clarification by the authors.

**Essential References Not Discussed:**

I cannot think of specific citations that I would deem essential to have discussed.

However, I believe that the paper would benefit from considering some of the recent literature on model multiplicity and fairness. While not requiring a specific citation, this literature would add some richness to the paper. Here are some examples-

D-Hacking (https://dl.acm.org/doi/abs/10.1145/3630106.3658928)
The Legal Duty to Search for Less Discriminatory Algorithms (https://arxiv.org/abs/2406.06817)
Fundamental Limits in the Search for Less Discriminatory Algorithms—and How to Avoid Them (https://arxiv.org/abs/2412.18138)
Operationalizing the Search for Less Discriminatory Alternatives in Fair Lending (https://dl.acm.org/doi/abs/10.1145/3630106.3658912)

**Experimental Designs Or Analyses:**

The experiment design as reported in the paper is robust and convincing. I particularly appreciated the development of a custom AutoML solution for multi-objective optimization. The cherry-picking examples would be sufficient to conceptually demonstrate x-hacking but the directed-search example shows that this potentially can be done efficiently and at scale.

**Methods And Evaluation Criteria:**

The proposed evaluation approach generally makes sense for illustrating the feasibility of “X-hacking.” The authors' choice of real-world datasets is appropriate, and their use of both off-the-shelf AutoML and a custom Bayesian optimization approach to systematically explore model multiplicity is reasonable and insightful.

However, (to somewhat repeat the point made above) the evaluation criteria focus primarily on the speed and ease of finding models that yield desired SHAP explanations relative to predictive performance. A critical limitation is that the authors do not clearly justify why the selective search for particular SHAP values constitutes harmful “hacking,” rather than legitimate model selection. The evaluation would benefit from explicitly distinguishing harmful selective practices from legitimate uses of model multiplicity, particularly by providing clear normative criteria or guidelines for differentiating the two scenarios.

**Other Comments Or Suggestions:**

see above

**Other Strengths And Weaknesses:**

Overall, I thought this was a great paper making a very important contribution to the literature. I can see this paper being very influential and opening up a future line of research.

**Questions For Authors:**

You convincingly demonstrate how easily AutoML tools can find models with desired SHAP explanations, but it's unclear precisely when selecting such a model becomes “manipulative” rather than beneficial or compliant (e.g., choosing models that intentionally exclude sensitive characteristics). Could you clarify what criteria or guidelines you envision for distinguishing harmful X-hacking from legitimate and beneficial model selection based on SHAP values?  A clear response would strengthen the practical relevance and conceptual precision of your paper, particularly for readers concerned with regulatory or ethical compliance.

**Relation To Broader Scientific Literature:**

The paper correctly situates itself within the model multiplicity and explainability literature. Although these themes have been explored extensively in the literature, the paper presents a novel and important approach and framework for considering the robustness of XAI in the context of searching for desirable explanations.

**Theoretical Claims:**

The paper is primarily an experimental paper and does not cover theoretical claims.

---

> ### Author Rebuttal · Authors · 2025-03-31
>
> We thank the reviewer for their strong approval of the paper and the valuable feedback. Below we address the questions raised by the reviewer.
>
> “**Selecting a model because it uses some characteristics rather than others (as accurately represented by SHAP) is not inherently problematic...**”:
>
> Choosing a model x ∈ R(t, D) where R(t, D) is the Rashomon set because it meets certain ethical, policy or domain-driven requirements is not problematic. Even if x ∈ C(t, D) (models that change explanations), a confirmatory analysis is legitimate if it is reported transparently. By contrast, malicious X-hacking occurs when an actor withholds other valid models from R(t, D) and only presents a model from the set C(t, D) ∩ R(t, D) that supports a desired narrative, thereby concealing alternative explanations that might contradict the narrative.  Also pointed out by reviewer **Se87**, we appreciate this distinction and intend to put it earlier in the paper (perhaps Section 1) and then later discuss more in Section 7 (Discussion)
>
> “**Could you clarify what criteria or guidelines you envision for distinguishing harmful X-hacking from legitimate and beneficial model selection based on SHAP values?**”: Our vision is that effective countermeasures or additional requirements from a publication venue to make reporting more transparent can help identify or raise flags for X-hacking which may require further scrutiny. Specifically,
>
> 1. Transparent reporting of model choice, clearly mentioning that different models yield different explanations and clarification of why the chosen pipeline/model’s explanation is more appropriate.
>
> 2. Pre-specifying a research plan and adherence to it which can be later verified will limit the degrees of freedom that produced the reported model.
>
> 3. Justifying the choices of explanations in a selected model may help give valuable insights.
>
> 4. A well documented research journey which is open, reproducible, and consistent with methodological standards will help identify X-hacking.
>
> 5. Awareness about model multiplicity and X-hacking and the need for effective countermeasures in the research community.
>
> While the above points focus more on what researchers reviewers and peers can look for and raise concerns based on the non-availability of information that threatens transparency, a full-fledged automatic detector for X-hacking is an ideal vision for further development.

---

### Official Review · Reviewer_2gSL · 2025-03-14

**Overall Recommendation:** 5

**Summary:**

In this work the authors propose the concept of “X-hacking,” where scientists or ML service providers leverage the multiplicity of ML systems to provide misleading explanations despite maintaining performance.

**Claims And Evidence:**

The authors claim the potential for X-hacking in ML systems, especially AutoML pipelines where one can scalably provide candidate models with different explanation behavior. They provide thorough evidence for this case both in a post-hoc setting (generating many models with good accuracy, finding one with a desired explanation) and ad-hoc setting (jointly optimizing performance and explanation goals in an AutoML pipeline). Their empirical results tell a compelling narrative of the existence of manipulable explanations, potential ways they can manifest, and how this manipulation can be optimized for.

However, multiple times throughout the paper the authors hypothesize about adversaries and countermeasures in an abstract manner. They say (paraphrasing) an adversary “could also consider risk of getting caught as an objective,” or that detection could occur through pipeline analysis without any experimental exploration of these ideas. I appreciate the discussion and understand the need to discuss all potential risks and countermeasures, but these claims felt very abstract and took away from the rigor found elsewhere in the paper. I might consider moving some of this discussion to the appendix to improve the clarity and focus of your work, but this is a minor comment and I still found their discussion interesting.

**Essential References Not Discussed:**

The authors covered most important references in this work.

**Experimental Designs Or Analyses:**

The author’s experiments are sound, specifically both their cherry picking/post-hoc setups (5.1) and directed search setups (5.2) although I am not very familiar with the optimization method used. I thought the analysis was good although I am not sure why the authors so heavily explored the benefits of Bayesian optimization over random optimizing in 5.2. Intuitively, optimizing for these objectives should be better/faster . However, I am more interested in knowing the gains an adversary would get over the cherry-picking method – could they get a desired X-hacking effect with fewer model training runs?

I had a question about the setup for 5.3 using simulated data. You include a dataset with correlated data so that you can shift SHAP weight around at will to manipulate explanations, correct? Why did you describe the features as colinear? Was the data perfectly dependent or just correlated? Also, in Figure 8, are each of the points in the graph a different model and the same feature? It should be clear that each point represents a choice of model, and so selecting points along a fixed horizontal line allows us to change feature dependencies in our explanation while maintaining performance.

**Methods And Evaluation Criteria:**

The authors use a Bayesian optimization and random sampler for the ad-hoc setting of X-hacking. The method makes sense for their joint prediction and X-hacking objective formulation.

**Other Comments Or Suggestions:**

None

**Other Strengths And Weaknesses:**

None

**Questions For Authors:**

In Fig. 3, are the features ordered by baseline importance? It might be good to mention that somewhere.

**Relation To Broader Scientific Literature:**

This work provides a great complement to literature on the Rashomon set and multiplicity by asking the opposite question of picking a model with a “bad” explanation. Additionally it builds nicely upon past work on the brittleness of explanations.

**Theoretical Claims:**

The authors make no theoretical claims in the main body of the paper.

---

> ### Author Rebuttal · Authors · 2025-03-31
>
> We thank the reviewer for their thorough and supportive comments. Furthermore, we will incorporate your suggestions to further strengthen the exposition and clarify details around our adversarial concepts, notation, and experimental setups. Specifically:
>
> We agree that our discussions of adversarial objectives, risk tolerance, and detection were presented at a more conceptual level compared to the more rigorous empirical work. Our focus was to demonstrate and quantify the practicality of manipulating ML explanations, not to propose or experimentally validate a complete “risk model” of adversarial behaviour. In a revised version, we intend to shift parts of Section 6 (Detection and Prevention) to Section 7 (Discussion) to improve clarity.
>
> “**… not sure why the authors heavily explored the benefits of Bayesian optimization over random optimization. I am more interested in the gains an adversary would get over the cherry-picking method — i.e., fewer runs to achieve a desired X-hacking effect.**”: We wanted to show that a malicious or benign user targeting explanations can exploit a more systematic search. We also wanted to show that given a greater time budget, random sampling will not on average perform better than Bayesian optimization in finding a defensible model. We explore this in an empirical manner and conclude that in our experiments, Bayesian optimization was 3 times faster at random sampling in the ad-hoc X-hacking setting (Section 5.4 (“Time to First Defensible Model”)).
>
> “**… you describe the data as collinear. Was it perfectly dependent or just correlated? Also, in Figure 8, is each point a different model? It would help to clarify that each point is a distinct pipeline choice and that for a fixed performance we can shift feature dependencies as we like. In Fig. 3, are the features ordered by baseline importance? It should be mentioned explicitly.**”:
>
> **Collinearity**: We used both moderate and high correlation scenarios, but not perfectly dependent features. Here we use “collinearity” referring to imperfect collinearity rather than perfect collinearity: i.e. the relationship is nearly but not exactly linear. We will clarify that the data is not perfectly dependent, but “redundant enough” to shift how SHAP values are allocated among correlated predictors without significantly affecting accuracy.
>
> **Figure 8**: Yes, each dot on the plot corresponds to a different trained model/pipeline. In the revision, we will explicitly state that “each point represents one pipeline configuration with its resulting performance and SHAP score for that feature.”
>
> **Figure 3**: Yes, the features in Figure 3 are shown in descending order of mean absolute SHAP under our baseline model. We will add a sentence in the caption or text noting that “features are ordered based on their baseline SHAP importance,”

---

### Official Review · Reviewer_Se87 · 2025-03-14

**Overall Recommendation:** 3

**Summary:**

The paper notes that AutoML pipelines allow training of multiple ML models, including sets of models with 'defensible' performance.

Thus, AutoML tools make it easier for authors to cherry-pick models to fit preconceived notions, as embodied by explainability metrics.  The paper uses the Shapley value as its representative of these.

Examples show that the possibilities for doing so increase in the redundancy of the feature set: when e.g. $x_1$ and $x_2$ are highly correlated, models may give weight to either without compromising their predictive power, allowing either to have a large Shapley value (the standard collinearity problem of econometrics/statistics).

# update after rebuttal

Stands.

**Claims And Evidence:**

The claims are assessed on 23 datasets, as well as some simulated data with known ground truth.  This seems fine.

**Essential References Not Discussed:**

n/a

**Experimental Designs Or Analyses:**

I have not checked any code.

**Methods And Evaluation Criteria:**

I have no concerns with the methods or evaluation criteria.

**Other Comments Or Suggestions:**

1. in econometrics, an increasingly common technique for addressing $p$-hacking is advance filing of research plans.  How well does this handle $X$-hacking concerns?
1. the $\mathcal{O}$ notation is used twice, once for 'obviousness' (p.3) and once for the 'risk of getting caught' (p.4).  Are these the same?  Overall, this does not make sense, as we do not have a model of cheating/getting caught.  The notation tends to be reserved for complexity measures, so is misleading.
1. other notation also seems inconsistent/confusing: e.g. $Q_D(m) = perf(m)$ and $Q_D(m) = \mathcal{I}(m)$.  I recommend picking one or the other convention.
1. I don't think that the paper needs to be cast in terms of unscrupulous authors: authors under time pressure, inexperienced authors, etc. will all cherry pick, wittingly or otherwise.
1. cut the 'audacity' section: you don't develop the idea; it feels like a distraction.
1. a fun experiment to establish $X$-hacking risks could be to: generate a defensible set with e.g. each feature given the largest Shapley value; ask an LLM to write an abstract for a paper corresponding to each.

**Other Strengths And Weaknesses:**

Overall, I found the paper's results unsurprising: more dimensions allow more misrepresentation.  This said, there needs to be a paper of record clearly establishing this.

I would _like_ this to be that paper.  To do so, though, the story and exposition need to be much tighter.  For example, I would like to see:
1. the set of 'defensible' models more tightly related to the Rashomon set: if they're the same, let's not duplicate language.
1. a more careful consideration of what datasets are more/less vulnerable to $X$-hacking: how do you explain the ranking of datasets in Figure 2?  Why does the ranking differ from that in Table 1?
1. it would be good to start the article with strong motivating examples 'in the wild' (p.8): at present, the concern is only plausible.

**Questions For Authors:**

See above

**Relation To Broader Scientific Literature:**

The paper is motivated by reference to $p$-hacking, from the statistics literature.

In the light of that literature, the present results are unsurprising: larger models offer more possibilities for misrepresentation - whether intentional or otherwise.  (My favourite example remains Bennett et. al.'s fMRI analysis of Atlantic salmon, winning him an IgNobel.)

I _think_ that slightly different issues arise: in the case of $p$-hacking, the problems arise from scale and 'chance'; in $X$-hacking, they arise from collinearity.  Is this correct?  If not, what is a more careful intuition?

It could also be useful to connect this to Wang, Rudin et al.'s 2022 TimberTrek, a visualisation tool for exploring Rashomon sets.

**Theoretical Claims:**

The paper presents no theoretical results, as such.

---

> ### Author Rebuttal · Authors · 2025-03-31
>
> We thank the reviewer for their valuable feedback. The following addresses the reviewer’s questions and concerns.
>
> **p-hacking and X-hacking**: Both p-hacking and X-hacking can arise from scale, chance and collinearity, which are interrelated rather than distinct. For instance, collinearity can cause regression estimates to become unstable, thus exacerbating p-value non-robustness. X-hacking, facilitated by AutoML, leverages model multiplicity and feature redundancy to align with desired explanations, also affected by these factors. X-hacking can be viewed as a generalization of p-hacking, extending the manipulation of statistical significance to the broader manipulation of model explanations; however the focus in our paper is on model-agnostic explanations, not including p-values, which tend to be considered only for a subset of models such as GLMs. Both phenomena can also occur unintentionally, highlighting the need for robust statistical practices and transparent reporting to mitigate these risks.
>
> **TimberTrek**: As pointed out by Reviewer 9ZtB also, in a revision we will cite TimberTrek in Section 2 as an important work on visualising and understanding Rashomon sets.
>
> **Rashomon set and defensible models**:  The Rashomon set R is defined as the set of models that exhibit nearly equivalent predictive performance within an acceptable threshold $\epsilon$.
>
> R = {$\{ m \in M | perf(m) \geq perf(m^*) - \epsilon \}$}
>
> Given M as the set of all possible models, perf(m) as the predictive performance of model m, and m∗ as the best-performing model, the set of "defensible" models is equivalent to the Rashomon set R. This set allows for a performance decrease ϵ(m) that may vary by model, accommodating those considered 'acceptable' due to their adherence to standard practices in the domain of application/publication. Therefore, the set of defensible models is a Rashomon set: R(t, D) in the paper (section 4.1). Additionally, the set C(t, D) ∩ R(t, D) is the set of those defensible models that changed the explanations. We will make it clearer and keep notation for the set C(t, D) ∩ R(t, D) (Pg. 2 para. 1) where we mention finding defensible models which are the models from the set C(t, D) ∩ R(t, D), i.e., “defensible models that changed explanations”, to avoid confusion.
>
> **Figure 2 and Table 1**: Figure 2 and Table 1 measure different aspects of the same set, C(t, D) ∩ R(t, D) defined in section 4.1. In figure 2, we see how many (the size) models in this set are found in a post-hoc manner by an off-the-shelf AutoML solution demonstrating that—with no special adjustments—AutoML is susceptible to X-hacking. On the other hand, in Table 1 we see how quickly the first member of this set is found by ad-hoc X-hacking. For a dataset the set C(t, D) ∩ R(t, D) may be large, yet a member of this set appears relatively late in the search process (longer time in Table 1). For another dataset C(t, D) ∩ R(t, D) may be small, but yields one member quickly during directed X-hacking (shorter time in Table 1).
>
> **A motivating example**:  Our following empirical analysis can be added as motivation after background section.
> Studies have shown a strong relationship between gender and cardiac disease [[98](https://shorturl.at/qn04f), [99](https://shorturl.at/fTX5J), [100](https://shorturl.at/vP9fo), [101](https://shorturl.at/Y8a0Z)], but in an empirical experiment with our ad-hoc X-hacking, the importance of the feature gender was manipulated to drop to Rank 6 from a Rank of 1 in a baseline. The set C(t, D) ∩ R(t, D) had many candidates. First such candidate was found only in 24 seconds.
> [This figure shows the results](https://imgur.com/TcEvbK9).
>
> **Pre-registration and X-hacking**:  We mention study pre-registration in Section 2 under p-hacking. Pre-registration is a valuable safeguard against X-hacking but might require more rigorous methodological detail. We will add this in Discussion.
>
> **Concern regarding notation**: Obviousness and risk of getting caught are both represented by O. It is an arbitrary function in the current context. To avoid confusion with complexity notation, we will change O to another letter, say Z.
>
> In Section 3 para. 3, we mention Q as any quality measure, which can be performance of a model perf(m) or an inferential summary of feature of interest I(m, x). It is only used to define a quality measure and later says (Eq. 1) that generally one optimises for a quality measure of performance during a model search: Q = perf(m).
>
> **Balanced view of X-hacking**: In Section 3 para. 2 we mention that such hacking can be “deliberate or not”. However, explicitly mentioning it earlier in Section 1 para. 2 can set a broader view at possible reasons behind X-hacking. Specifically, we will add the phrase “unscrupulously, or through lack of time or experience”.
>
> **Editing “audacity” section**: To maintain the focus of the reader, the subsection 3.3 will be removed to supplementary materials.

---

> > ### Comment · Reviewer_Se87 · 2025-04-04
> >
> > Thanks!
> >
> > Using your categories:
> >
> > **p- and X-hacking**: thanks.  I've lost track of whether I can see the revised version in ICML, but would like to see that.
> >
> > ** TimberTrek**: thanks.
> >
> > **Rashomon sets**: oh good!  I'd strongly encourage you to fully adopt the Rashomon terminology: maintaining terminological consistency helps maintain clarity in the field.
> >
> > **Fig 2, Table 1**: thanks.  I'd find valuable intuitions for _why_ $C(t, D) ∩ R(t, D)$ be large, but members of it slow to appear, and _vice versa_.
> >
> > **Motivating example**: to clarify, by 'in the wild', I meant an example where you think that existing results have been X-hacked.  The gender/cardiac example goes in a different direction: if we believe that the consensus in the existing published literature is correct (even though we can manipulate models to get other results), then it doesn't serve as an example of X-hacking breaking out into the wild.
> >
> > **pre-registration**: thanks!
> >
> > **notation**: as originally mentioned, you seem to wave your hands at an unspecified game-theoretical model.  I saw this as muddying waters rather than adding value, so recommend cutting this as much as possible, to free space to properly exposit core points.
> >
> > re: $perf$ and $\mathcal{I}$, I'd found it ugly and confusing to mix notational systems (is the latter a set?) for similar concepts.
> >
> > **balanced view**: thank you; I'd recommend pruning all other references to motivation - for whatever reason, this can happen.
> >
> > **audacity**: I didn't see the point of this material at all; I don't think that the appendices should be storage cupboards for questionable material; 'kill your darlings', editors say.
> >
> > % do I need more?

---

> > > ### Author Response · Authors · 2025-04-07
> > >
> > > We address the reviewer's comments below.
> > >
> > > **Rashomon sets**: to maintain clarity in the field, we will fully adopt the Rashomon terminology.
> > >
> > >
> > > **Intuitions for size of C(t, D) ∩ R(t, D)**: intuitively, there can be a number of reasons for why C(t, D) ∩ R(t, D), be large, but members of it slow to appear, and vice versa.
> > >
> > > 1. Behaviour of the search algorithm: an optimiser may initially explore diverse hyperparameter settings only later narrowing on certain promising regions in a large pipeline search space given by an AutoML solution.
> > >
> > > 2. Complexity of the running pipeline: pipelines having complex multiple steps may slow down how rapidly one can iterate to find the one that belongs to C(t, D) ∩ R(t, D).
> > >
> > > 3. Size of the dataset: large number of features slows down the calculation of SHAP values as SHAP computations grow exponentially with feature count. Moreover, pipeline evaluation on a larger dataset may take more time.
> > >
> > > 4. Information redundancy among features: in a dataset with several correlated features the set C(t, D) ∩ R(t, D) may be large but the search might not systematically explore these dimensions until later.
> > >
> > >
> > >
> > > **“In-the-wild" example**:
> > >
> > > An ideal point of reference would be a secondary source, i.e. a published report levelling criticism against a study whose AI-derived explanations or insights were found not to be robust to their upstream modelling decisions, such as in a commentary article, a retraction or coverage on [RetractionWatch](https://retractionwatch.com/) . However, we were unable to find such an example that specifically highlights the use of either XAI or AutoML; of course many such examples exist for p-hacking and poor research more generally. Absent such a secondary source, the alternative is to find primary sources, i.e. examples of papers or preprints where we are able to determine, with independent access to the same dataset, the non-robustness of the published results, or where the authors inadvertently reveal the same through their reporting. However, a systematic search for such primary sources would be an involved process and certainly beyond the scope of the current paper. Such an audit or meta-analysis might be a good area for future research.
> > >
> > > We demonstrate that there is both means and motive for X-hacking. For example, prior studies show how to deliberately alter a model’s behaviour to produce a preferred explanation [[1](https://doi.org/10.48550/arXiv.1911.02508)]. In contrast, X-hacking achieves a similar effect without modifying the underlying model, by systematically searching a large pipeline space for explanations that fit a given agenda. Whether pursued deliberately or not, X-hacking is relatively easy to perform, underscoring its practical feasibility—and the importance of awareness in the research community.
> > >
> > >
> > >
> > > **Notation**: we will keep the notation to QD (m) = perf(m) and omit I(m, x) to be defined by Q.
> > >
> > >
> > >
> > > **Audacity**: We agree with your suggestion and remove the ‘audacity’ section to streamline our argument and keep the paper focused.

---

### Official Review · Reviewer_9ZtB · 2025-03-25

**Overall Recommendation:** 3

**Summary:**

The paper introduces X-hacking that manipulates XAI metrics by exploiting model multiplicity to find explanations supporting desired conclusions. Bayesian optimization helps to find models that support a desired explanation while maintaining acceptable accuracy, allowing for manipulation of SHAP values.  Datasets with high feature redundancy are particularly vulnerable to X-hacking. The paper also discusses detection and prevention strategies, highlighting ethical concerns and the risks X-hacking poses.

## update after rebuttal
Thank you to the authors for their reply and willingness to further improve the manuscript. I would like to encourage the authors to incorporate a discussion on ethical issues related to the intent of model selection under multiplicity. While I still believe that the paper would greatly benefit from additional results based on non-feature-attribution explainability methods, I acknowledge that the empirical results for AutoML pipelines are extensive. I have re-read the paper and the authors’ reply and have adjusted my scores accordingly.

**Claims And Evidence:**

Claim 1: Automated machine learning (AutoML) pipelines can be easily adapted to exploit model multiplicity at scale, making them vulnerable to X-hacking.

Claim 2: Bayesian optimization accelerates X-hacking for features susceptible to it, compared to random sampling.

Claim 3: The paper shows that some datasets are more vulnerable than the others to X-hacking (probably due to different level of multiplicity).

**Essential References Not Discussed:**

The field has a lot of recent papers, and the authors did a great job referencing important papers. However, some literature on the Rashomon set is missing. See [Ganesh, The Curious Case of Arbitrariness in Machine Learning, 25] for reference.

**Experimental Designs Or Analyses:**

The paper's experimental designs is generally sound for demonstrating X-hacking.

**Methods And Evaluation Criteria:**

The methods presented in the paper effectively illustrate the concept of X-hacking and the role of AutoML. However, they can be time-consuming and likely will be hard to generalize to more complex models.

**Other Comments Or Suggestions:**

NA

**Other Strengths And Weaknesses:**

**Strength**
The analysis of different features to understand their susceptibility to X-hacking is particularly interesting

**Weaknesses**

More nuanced view on X-hacking is needed, as it can easily be used as an assistive tool to align model with the domain experts intuition.

Only one XAI methos is used, which is feature importance-based metric, so it is not clean how X-hacking behaves for other XAI methods.

**Questions For Authors:**

How does the paper results generalize to other explainability methods that are not feature importance based?

**Relation To Broader Scientific Literature:**

The core idea that model multiplicity allows selecting models with different feature importances (Shapley values) is not new. It's well known that different models in the Rashomon set can emphasize different features while achieving similar predictive accuracy (see Rudin Amazing Things Come From Having Many Good Models, 24). Therefore one can choose models based on these features. There are tools that visualize these models as well (Timbertrek, 22).

**Theoretical Claims:**

The paper does not make theoretical claims.

---

> ### Author Rebuttal · Authors · 2025-03-31
>
> We thank the reviewer for their thoughtful feedback and valuable references.
>
> We agree that the phenomenon of having many equally accurate, yet interpretively distinct, models is well established, and we cite several related papers (e.g., Fisher et al., 2019; Brunet et al., 2022) to acknowledge this. However, our paper contributes a new perspective by explicitly demonstrating how modern AutoML pipelines can be adapted (or misused) to exploit those known multiplicities at scale to manipulate XAI metrics (referred to as “X-hacking”). While multiplicity has certainly been studied, we highlight in the related work section how this is limited to certain model families. Our approach highlights automation and practicality—showing that even modest budgets given to AutoML systems can allow for systematically “cherry-picking” explanations to support a predefined narrative, all while preserving acceptable predictive performance. This practical demonstration of how easily multiplicity can be exploited via off-the-shelf or lightly customized AutoML solutions distinguishes our work from purely theoretical or smaller-scale explorations of Rashomon sets. We acknowledge the recent preprint, Ganesh (2025) and add a reference to this in Section 2 (Background).
>
>
> “**More nuanced view on X-hacking is needed, as it can be used as an assistive tool to align the model with domain experts’ intuition.**”: We acknowledge that choosing models that align with intuitions of domain experts may not always raise ethical concerns. For example, when domain experts prefer interpretable patterns that reflect established science or policy. However, the distinction lies in intent and transparency. We appreciate this insight and intend to add in Section 7 (Discussion) that model selection for domain alignment may not always attribute to malpractice provided it is transparent and reported in good faith. While our paper emphasizes the opposite, an expanded discussion of both benign and malicious scenarios will enhance the paper’s balance.
>
>
> “**Only one XAI method is used … it is not clear how X-hacking behaves for other XAI methods.**”: We worked with SHAP for two reasons: it is a model agnostic approach, and it is a popular metric, making it a strong representative for our demonstration. We show that quantitative XAI metrics like SHAP can be easily manipulated using off-the-shelf AutoML solutions, thus sufficiently establishing the core risk of X-hacking at scale. We believe that any post-hoc explanation method -- especially those susceptible to model multiplicity -- will face similar risks, however, systematic exploration of other XAI metrics is planned as future work.
>
>
> “**The methods can be time-consuming and might be hard to generalize to more complex models.**”: We share the concern that scanning a massive analysis space can be computationally expensive, however, we show that current off-the-shelf AutoML solutions and their search spaces can be used to find “defensible” models even under modest resource budgets. For extremely large or complex models, the cost (computational resources and time) of repeated training might be higher, however, it is the choice of the researcher to allot higher budgets in the context of their research. Our current intention is to show that X-hacking can be easily and effectively done at scale, even with small budgets.

---

### Decision · Program_Chairs · 2025-05-01

**Decision:**

Accept (poster)

**Comment:**

The paper studies the manipulation of XAI metrics by exploiting model multiplicity to find explanations that support specific conclusions.

The reviewers generally liked the setting of the paper and found the experimental design to be convincing and robust. In particular, providing cherry-picking examples is by itself sufficient to demonstrate x-hacking. However, the authors take the additional to show that by directed-search, the x-hacking can be done efficiently at scale.

There were minor concerns about missed references and a focus only on SHAP values/feature importance metrics. I encourage the authors to cite the provided references and add a discussion about the generalizability of their results to other explainability techniques in the limitation/future work section.